# Cholinergic basal forebrain nucleus of Meynert regulates chronic pain-like behavior via modulation of the prelimbic cortex

Manfred J. Oswald [1], Yechao Han [1], Han Li[1], Samuel Marashli [1], Deniz Nouri Oglo[1], Bhavya Ojha [1], Paul V. Naser [1], Zheng Gan[1] & Rohini Kuner [1] ✉

The basal nucleus of Meynert (NBM) subserves critically important functions in attention, arousal and cognition via its profound modulation of neocortical activity and is emerging as a key target in Alzheimer's and Parkinson's dementias. Despite the crucial role of neocortical domains in pain perception, however, the NBM has not been studied in models of chronic pain. Here, using in vivo tetrode recordings in behaving mice, we report that beta and gamma oscillatory activity is evoked in the NBM by noxious stimuli and is facilitated at peak inflammatory pain-like behavior. Optogenetic and chemogenetic cell-specific, reversible manipulations of NBM cholinergic-GABAergic neurons reveal their role in endogenous control of nociceptive hypersensitivity, which are manifest via projections to the prelimbic cortex, resulting in layer 5-mediated antinociception. Our data unravel the importance of the NBM in top-down control of neocortical processing of pain-like behavior.

A major hindrance to adequate therapy of chronic pain disorders is given by incomplete knowledge on brain circuits underlying the perception of pain and their modulation over the transition from acute to chronic pain. Their elucidation is therefore important for yielding mechanistic insights as well as for therapeutic advance. Recent studies on functional interrogation of brain circuits have led to breakthroughs on structure-function properties of some brain networks involved in pain, and particularly revealed key roles for neocortical domains[1]. Pain perception is subject to profound modulation by contextual, environmental, and psychosocial factors. As contributing neural mechanisms, insights are now emerging on modulation of neocortical processing by afferent input from GABAergic, dopaminergic, and serotonergic pathways[2].

In comparison, very little is known about the scope and functions of cholinergic pathways in the brain in modulating pain perception. This is in contrast to extensive pharmacological studies from the past two decades, which report effects of cholinergic signaling via both ionotropic nicotinic receptors as well as metabotropic muscarinic receptors on pain and analgesia[3]. Systemic, peripheral, as well as spinal

administration of cholinergic ligands modulates nociception and studies with central administration have implicated cholinergic signaling in opioidergic analgesia and descending modulatory systems. However, there has been very little progress in exploiting cholinergic modulation towards pain relief, owing primarily to major gaps in understanding the underlying circuitry, particularly with respect to the delineation of the origin of cholinergic inputs. This is particularly important because both facilitatory and inhibitory effects are associated with pharmacological modulation of cholinergic receptors, which can be attributed not only to diversity of receptor-mediated signaling but also to the locus of cholinergic modulation in the nervous system.

In the brain, cholinergic neurons are abundant either in form of local interneurons in specific areas, such as the caudate putamen, or organized in the cholinergic nuclei Ch1–Ch6 of the basal forebrain and brainstem to function as projection neurons with distant targets[4]. Amongst these, the basal forebrain system comprises discrete groups of cholinergic cells (Ch1–Ch4), with neurons in the medial septum (MS) and the vertical limb of the diagonal band of Broca (vDB) primarily

[1]Pharmacology Institute, Medical Faculty Heidelberg, Heidelberg University, Im Neuenheimer Feld 366, 69120 Heidelberg, Germany.
✉ e-mail: rohini.kuner@pharma.uni-heidelberg.de

targeting the hippocampus, while the neurons in Ch4 largely account for the cholinergic input into the neocortical mantle and also project to the amygdala[4]. In the rodent brain, the structure most analogous to the Ch4 is given by the nucleus basalis magnocellularis (NBM; basal nucleus of Meynert), also extending into a band ventral to the anterior commissura called the substantia innominata, which are collectively referred to under the term NBM in this study, consistent with several other published studies (e.g., ref. 5; schematic view in Fig. 1a). This sector contains the largest component of corticopetal projections from the basal forebrain and is overwhelmingly cholinergic in nature. The NBM has been ascribed a modulatory role in specific key functions, such as arousal, attention, fear, and social interactions, including social recognition memory[5]. Furthermore, the NBM has been implicated in sharpening the acuity of sensory processing by enhancing the "signal-to-noise" ratio in cortical circuits via nicotinic and muscarinic mechanisms involving both pyramidal neurons and GABAergic interneurons[6,7]. These properties potentially place the NBM in a critical position to modulate pain perception and its plasticity, given the importance of neocortical processing in pain[1]. Surprisingly, however, the NBM has hardly been studied in the context of pain, barring a few studies with excitotoxic lesions and broad toxin-mediated ablation of cholinergic groups. Importantly, there have been no studies functionally delineating the underlying native circuitry. Moreover, it remains unknown whether and how activity patterns in the NBM change in association with pain and the NBM undergoes plasticity during the transition to chronic pain in vivo.

Here, we performed in vivo recordings using tetrodes to dynamically capture changes in activity of single neurons as well as oscillatory field rhythms in the NBM in freely-moving, behaving mice during nociception and the transition to inflammatory hypersensitivity. We report specific responses of the NBM to pain-inducing (noxious) stimuli, which demonstrate a switch in responsivity to low-intensity stimuli in an inflammatory pain model, thus mirroring behavioral hypersensitivity. Simultaneously, gamma and beta oscillatory rhythms undergo potentiation of spectral power. Using reversible, cell type-specific chemogenetic and optogenetic manipulations in conjunction with behavior, we demonstrate that this potentiation of cholinergic activity in the NBM and its projections to the prefrontal cortex suppresses nociceptive hypersensitivity in both inflammatory and neuropathic pain conditions, thus paving the way for therapeutic strategies specifically targeting these cholinergic cell groups.

## Results

### Oscillatory rhythmic activity in the NBM in nociception and inflammatory pain

Intraplantar hindlimb injection of capsaicin in wild-type mice, which acutely induces strong, tonic pain, led to a significant increase in expression of the activity-dependent immediate early gene product, Fos, in the NBM (schematically shown in Fig. 1a), including cholinergic neurons as seen via co-labeling for the marker choline acetyltransferase (ChAT; Fig. 1b). We next targeted this area in electrophysiology experiments in awake, behaving mice to directly study changes in NBM activity, both at the level of field potentials and single cells using tetrodes. Application of mechanical force via von Frey filaments was associated with an increase in activity across all frequency bands over baseline (pre-stimulus) activity levels (Fig. 1d, e; also see Supplementary Fig. 1a). Across multiple trials and animals, the increase was statistically significant in the power of beta oscillations (14–30 Hz) when stimuli at and above the nociceptive thresholds were applied (von Frey force of 0.6–1.0 g) as well as with low-intensity, non-noxious tactile stimuli (0.07–0.16 g), whereas power of gamma oscillations (30–100 Hz) selectively increased with nociceptive strength stimulation (Fig. 1f). This finding is particularly interesting because gamma oscillations in cortical areas have been functionally linked with nociception in both human

and rodent studies[8–10], and are known to be associated with synchronization of activity via GABAergic interneurons[11]. Noxious mechanical stimulation-induced increase in beta activity as well as gamma activity reached statistically significant levels prior to the behavioral nocifensive response and was maintained for 2 s after application of the stimulus (Fig. 1g, for comparison, data on non-noxious stimulation are shown in Supplementary Fig. 2b).

We next sought to test the potential significance of the NBM in the progression of nociception to hypersensitivity that is characteristic to persistent inflammatory pain. Indeed, mice with unilateral hindpaw inflammation induced by injection of Complete Freund's Adjuvant (CFA), which demonstrate nociceptive hypersensitivity (Supplementary Fig. 1c), demonstrated enhanced Fos expression in cholinergic neurons in the NBM (Fig. 2a, b). We then compared oscillatory activity between naive conditions and after CFA-induced hypersensitivity was established. At 24 h after CFA injection in the hindpaw, which corresponds to the time at which behavioral hypersensitivity to mechanical stimulation reaches a peak in our hands, paw stimulation elicited a significantly larger increase in the power of beta and gamma rhythms (Fig. 2c, d), but not of alpha and theta activity (Supplementary Fig. 1d) in the NBM. An interesting finding was that the inflammatory pain-associated increase in gamma and beta power was seen with low intensities of mechanical stimulation, which are typically non-noxious in physiological conditions but are perceived as noxious in the inflamed state (Fig. 2e). Taken together, these findings indicate that the NBM is recruited during nociception and shows facilitation of its responsivity over the transition to hypersensitivity in inflammatory pain-like behavior.

### Single-cell analysis of NBM activity in nociception and inflammatory pain

Analyzing activity at the single-cell level via spike sorting led to interesting insights into the cellular nature of NBM responsivity and plasticity in pain. Amongst the 221 units recorded under naive conditions, less than 10% showed a consistent increase or a decrease in firing rate in withdrawal trials upon applying 20 mechanical paw stimulations with the weak or strong filament (Fig. 3a, b); Example traces and average Z-scores (denoting number of standard deviations for data points above or below mean) are shown in Fig. 3a and unit proportions in Fig. 3b. In mice with inflammatory pain-like behavior, the proportion of units responding to mechanical stimulation did not change significantly during strong hypersensitivity over the first 4 days after CFA injection (Fig. 3b and Supplementary Fig. 2a). Over this period however, maximal z-score values increased significantly in neurons excited by noxious intensities of mechanical stimulation (Fig. 3c), but not in neurons inhibited by mechanical stimulation (Supplementary Fig. 2b), thus corresponding to the overall increase in the power of oscillatory activity which we observed at the LFP level. Furthermore, by analyzing the shape of the spike waveform, we then classified units into Class 1 (Fig. 3d) and Class 2 (Fig. 3e) with broad or narrow spike wave-forms, respectively[12]; fast-spiking classes of GABAergic projection neurons and interneurons are represented within class 2 units[13]. Interestingly, only class 2 neurons showed a statistically significant increase in activity in response to sensory stimulation in mice with paw inflammation as compared to control mice (Fig. 3d, e). These data suggest that NBM neurons, which are excited by mechanical stimuli, undergo facilitation over the manifestation of inflammatory nociceptive hypersensitivity and further that fast-spiking, class 2 GABAergic neurons in the NBM particularly contribute to these changes. This finding is noteworthy, since in the mouse NBM, 92% of ChAT-expressing cholinergic neurons are known to be GABAergic[14]. At late time points after CFA injection (7–14 days), after normal nociceptive sensitivity is recovered, we observed that patterns of oscillatory and single-cell activity in the NBM not only normalize, but partly even fall below baseline values (Supplementary Figs. 2b and 3a, b).

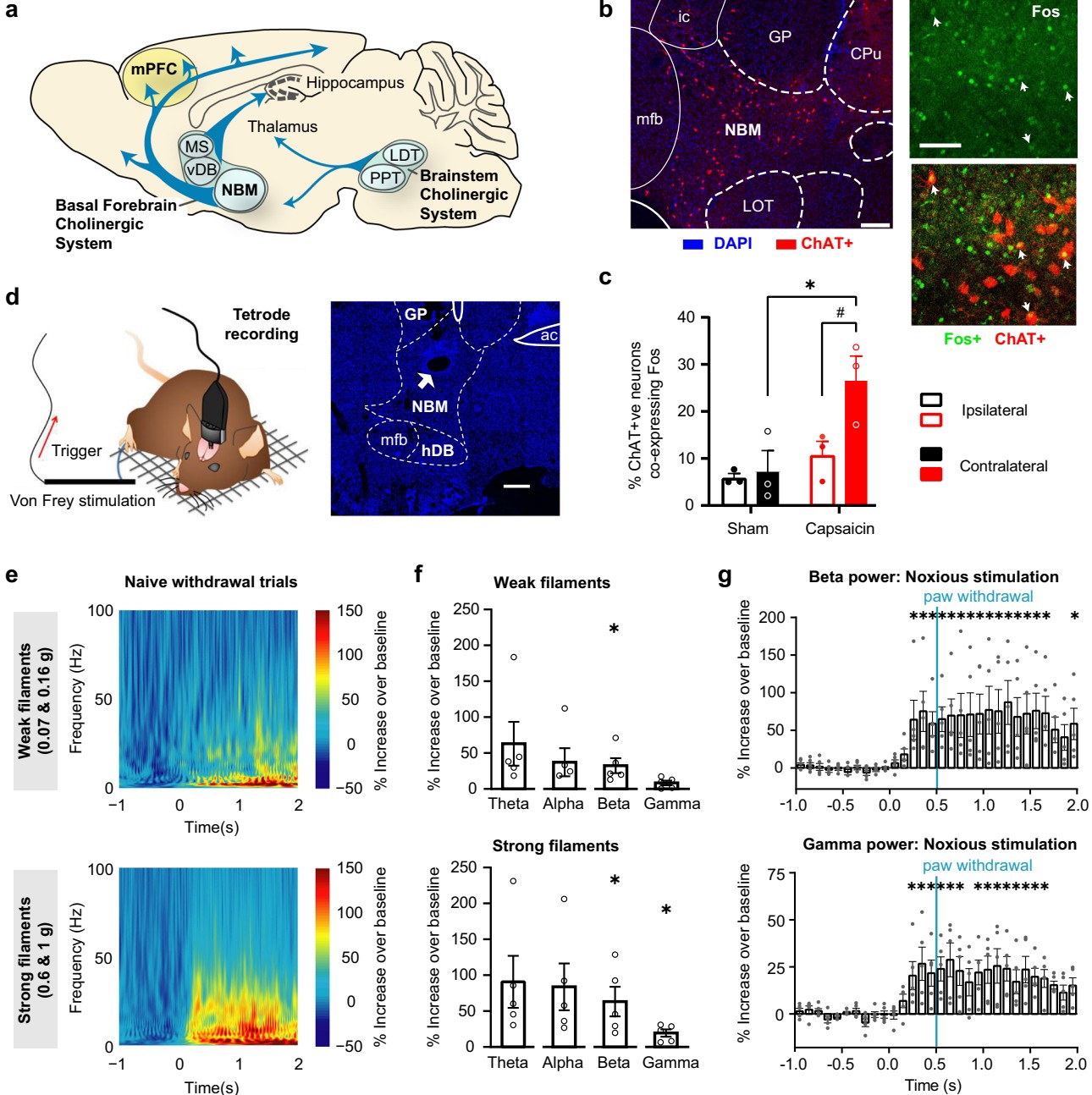

**Fig. 1 | Recruitment of the Nucleus Basalis of Meynert (NBM), a basal forebrain cholinergic nucleus, by noxious stimuli eliciting pain-like behavior.**
**a** Schematic of the main cholinergic nuclei in mouse brain (**b**, **c**) Typical examples (**a**) and quantification (**b**) of capsaicin-induced increase in Fos immunohistochemistry in NBM cholinergic neurons (ChAT + ve; arrows: co-labeled cells). $n = 3$ mice/group; $P < 0.05$ (*0.0103, #0.0263), two-way ANOVA with Sidak's multiple comparisons test. **d** Schematic representation (sourced from ref. 31) of in vivo tetrode recordings in the NBM (white arrow: electrode tip lesion in section) in response to von Frey mechanical stimulation of hindpaw. **e** Mean time–frequency representation of spectral modulation in the NBM for all trials with paw-withdrawal response to either weak filaments (0.07 g and 0.16 g;) or strong filaments (0.6 g and 1 g). **f**, **g** Corresponding quantification of power (**f**) of oscillatory activity in frequency ranges theta (4–8 Hz), alpha (8–14 Hz), beta (14–30 Hz) and gamma (30–100 Hz), represented as % change in 2 s post-application period over 1 s baseline activity prior to stimulus application, and corresponding time course (**g**); vertical blue line: time of paw withdrawal. In **e**, **g**, $n = 5$ mice; *$P < 0.05$; One-sample $t$

test (two-tailed) was employed in **f**; $t$ and $P$ values for the different groups are as follows: Weak filaments: $t = 2.06, 1.89, 3.10, 2.60$; $P = 0.108, 0.132, 0.036, 0.060$; Strong filaments: $t = 2.50, 2.57, 3.07, 3.81$; $P = 0.067, 0.062, 0.037, 0.019$; for theta, alpha, beta & gamma, respectively. One-way repeated measures ANOVA with Dunnet's multi-comparison versus pre-stimulation baseline was employed in **g** (*$P = 0.0058, 0.0007, 0.0143, 0.0048, 0.0021, 0.0019, 0.0106, 0.0013, 0.0005, 0.0007, 0.0001, 0.0028, 0.0013, 0.0006, 0.0011, 0.0148$ for beta, and $0.0091, 0.0002, 0.0045, 0.0012, 0.0001, 0.0022, 0.0036, 0.0016, 0.0005, 0.0012, 0.0451, 0.0015, 0.0124, 0.0197$ for gamma power time bins, from left to right, respectively). Scale bars represent 0.5 mm and 50 µm (right) in **b** and 250 µm in **d**. MS medial septal nucleus, vDB diagonal band of Broca, LDT laterodorsal tegmental nucleus, PPT pedunculopontine tegmental nucleus, mPFC medial prefrontal cortex, ac anterior commissure, CPu caudate putamen, GP glopus pallidus, LOT nucleus of the lateral olfactory tract, mfb medial forebrain bundle, ic internal capsule, ERP event-related potential or perturbation. Data are presented as mean +/− standard error of the mean (SEM).

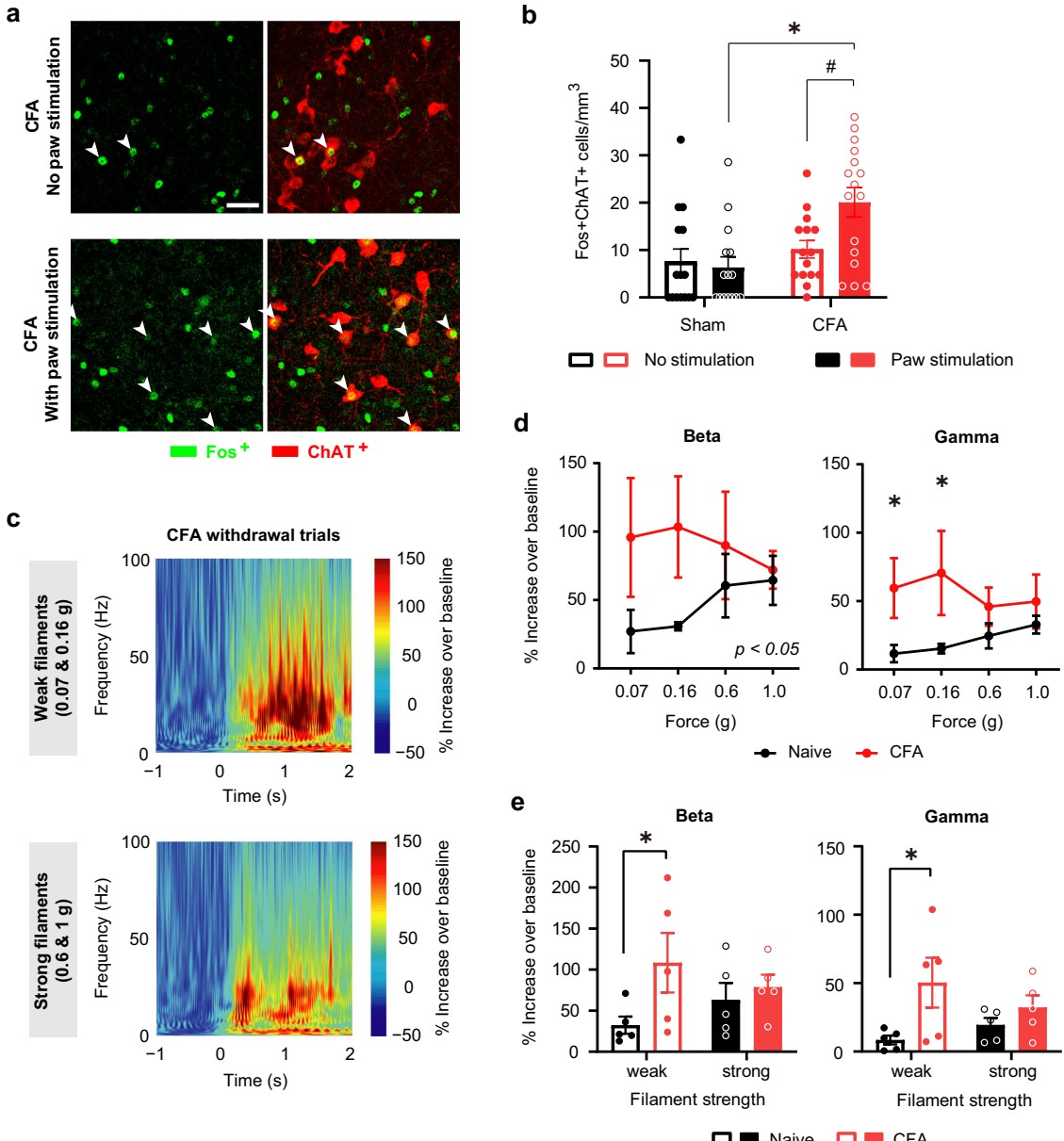

**Fig. 2 | Enhanced responsivity in cholinergic neurons and increased power of beta and gamma oscillatory activity in the NBM at peak of inflammatory pain-like behavior. a, b** Comparison of the activity of cholinergic neurons of the NBM in the presence or absence of application of mechanical stimulation with 0.16 g force to the contralateral plantar hindpaw under baseline conditions or 1 day after CFA injection. Shown are typical examples (**a**) and quantification (**b**); $n = 4$ mice/group; $P < 0.05$ (*0.0005, #0.0134), two-way ANOVA with Sidak's multiple comparisons test. **c** Time−frequency representation of spectral power in the NBM in mice at day 1 after CFA injection ($n = 5$ mice/treatment). **d, e** Comparison of the power of oscillatory activity in beta and gamma frequency ranges between naive (sham) conditions and CFA day 1, calculated as % increase in 2 s post-application period over 1 s baseline activity prior to stimulus application. Shown are stimulus−response curves (**d**) or analysis of % change in spectral power (**e**) in response to innocuous filaments (0.07 and 0.16 g) and noxious mechanical pressure (0.6–1.0 g); $n = 5$ mice/group; *$P < 0.05$ (0.0417 for 0.07 g, 0.0244 for 0.16 g in **d**; 0.0425 for beta-weak & 0.0095 for gamma-weak in **e**), two-way ANOVA with Sidak's multiple comparisons test. Data are presented as mean +/− SEM.

## Optogenetic stimulation of NBM acutely suppresses hypersensitivity

To directly uncover the significance of our findings, we employed a cell-specific optogenetic approach by targeting the blue light-gated cation channel Channelrhodopsin to ChAT-expressing neurons. Recombinant adeno-associated virions (AAV) were stereotactically injected to express yellow fluorescent protein-tagged Channelrhodopsin in a Cre-dependent manner (rAAV-Dio-ChR2-YFP) unilaterally in the NBM of ChAT-Cre transgenic mice (Fig. 4a, b). Cre-negative mice subjected to the same treatments served as controls. Delivery of blue light to the NBM via chronically implanted optic fibers significantly increased Fos expression in cholinergic neurons, thereby establishing in vivo validation of the approach (Fig. 4c, d). Upon blue light stimulation, baseline sensitivity to mechanical stimuli remained unchanged (Fig. 4e, baseline); however, the left-ward and up-ward shift in the von Frey stimulus−response function, representing the manifestation of inflammatory hypersensitivity, was significantly lowered in Cre+ mice as compared to Cre- controls when tested at peak sensitization on day 2 post-CFA (Fig. 4e, middle panel). Likewise, the mechanical withdrawal threshold was significantly increased in mice with CFA upon blue light stimulation in Cre+, but not in control Cre- mice (Fig. 4f). Overall, the magnitude of mechanical hypersensitivity over baseline

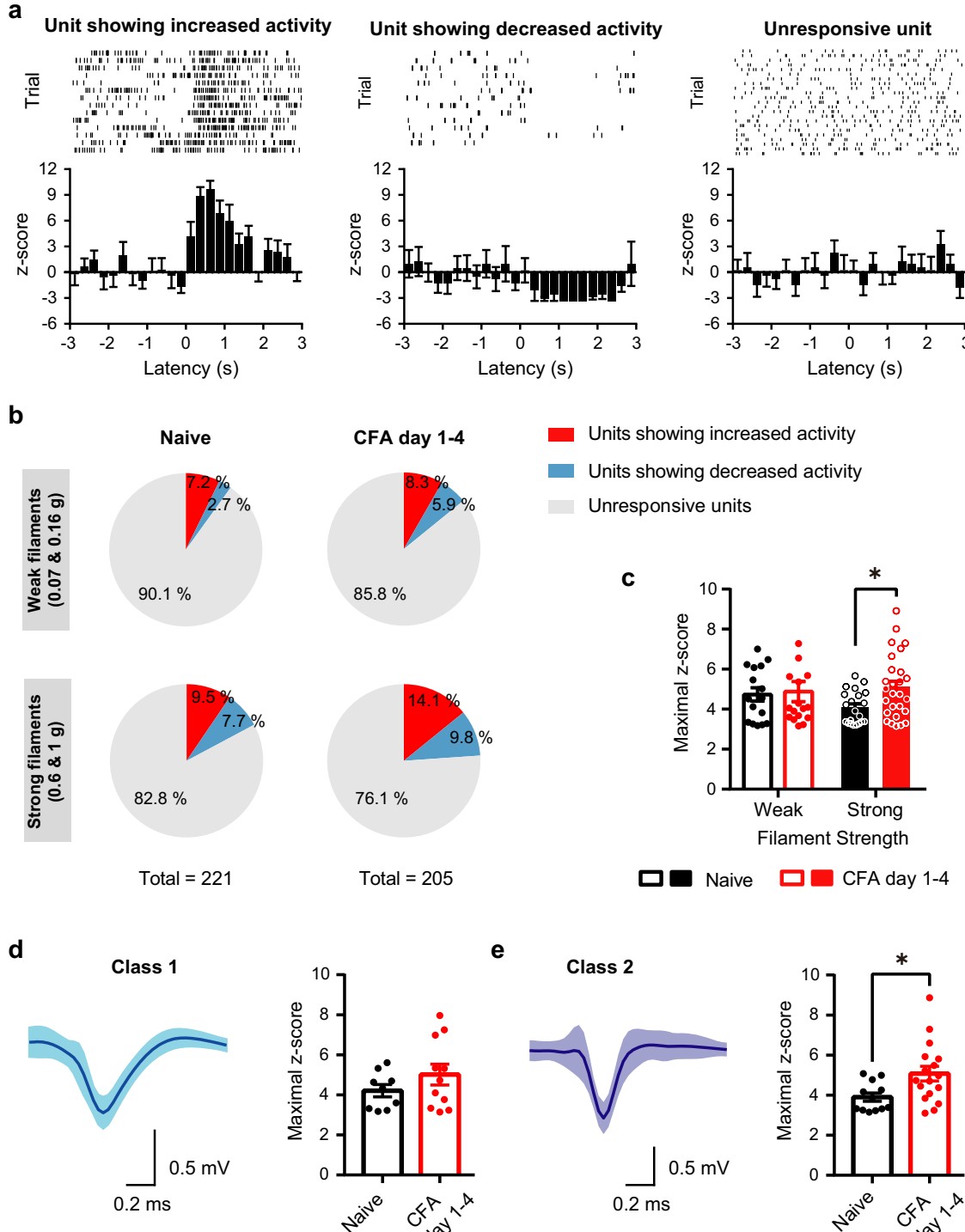

**Fig. 3 | Resolving changes in NBM activity during acute nociception and inflammatory pain-like behavior at the single-cell level. a** Typical examples (upper panels) and average z-scores, which represent the number of standard deviations for data points above or below mean, demonstrating NBM units that are excited by noxious stimulation of the paw (left-most panel), units that are suppressed in activity by paw stimulation (middle panel) and units that are not altered significantly in activity following paw stimulation. **b** Distribution of NBM units responding to mechanical stimulation in naive (sham) conditions and during hindpaw CFA-induced peak inflammatory pain-like behavior (day 1–4). **c**–**e** The maximum enhancement of activity over average baseline values for units excited by mechanical stimulation is demonstrated in naive mice and post-CFA. In **d**, **e**, units are subdivided into class 1 (**d**) and class 2 (fast spiking; **e**) types of neurons based on waveform. $n = 5$ mice/group; *$P < 0.05$ (0.0412 in **c**; 0.0191 in **e**), two-way ANOVA with Sidak's multiple comparisons test (**c**), and unpaired two-tailed $t$ test (**d**, **e**). Data are presented as mean +/− SEM.

values was decreased and the return to baseline sensitivity was faster upon optogenetic stimulation of NBM cholinergic neurons (Fig. 4f). However, CFA-induced heat hyperalgesia was not significantly altered in magnitude or duration (Fig. 4g).

## Dissecting the contribution of NBM cholinergic-GABAergic projections to the medial prefrontal cortex

Because the NBM projects to a large number of neocortical targets, many of which affect pain and hypersensitivity in multiple ways, we

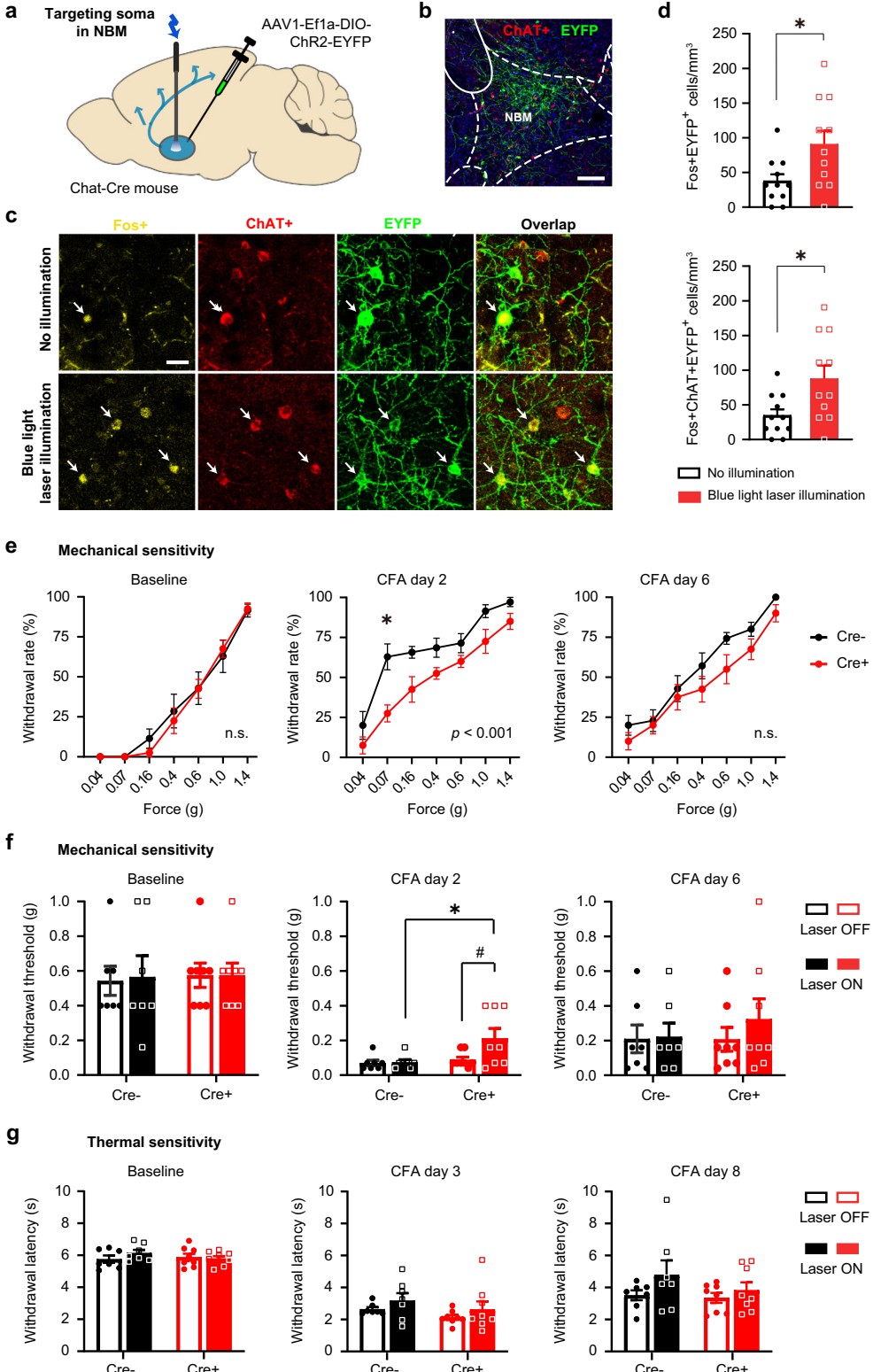

**Fig. 4 | Optogenetic activation of the NBM cholinergic neurons attenuates inflammatory mechanical, but not thermal, hypersensitivity. a** Scheme for optogenetically manipulating NBM cholinergic neurons with blue laser light. **b**–**d** Expression of the excitatory opsin, Channelrhodopsin-EYFP in the NBM (**b**), typical examples (**c**) and quantification (**d**) of enhanced Fos expression in EYFP+ and ChAT+ neurons (arrows in **c**) with blue light, thus validating the efficacy of optogenetic activation in vivo. $n = 3$ mice/group; *$P < 0.05$ (0.0234, top; 0.0183, bottom), unpaired two-tailed $t$ test. Scale bar = 200 μm in **b** and 25 μm in **c**.

**e**, **f** Significant attenuation of peak mechanical hypersensitivity (day 2) induced by intraplantar CFA injection, shown as stimulus–response curves (**e**) and withdrawal thresholds (**f**) in response to von Frey stimulation; $P$ values in inset represent ANOVA-based comparison of the two entire stimulus–response curves. **g** Lack of modulation of hypersensitivity to a heat ramp. **e**, **f** $n = 7$ ChAT-Cre- and 8 ChAT-Cre+ mice; $P < 0.05$ (*0.0275, middle in **e**; *0.0139, #0.0231, middle in **f**), two-way ANOVA with Sidak's multiple comparisons test. Data are presented as mean +/− SEM.

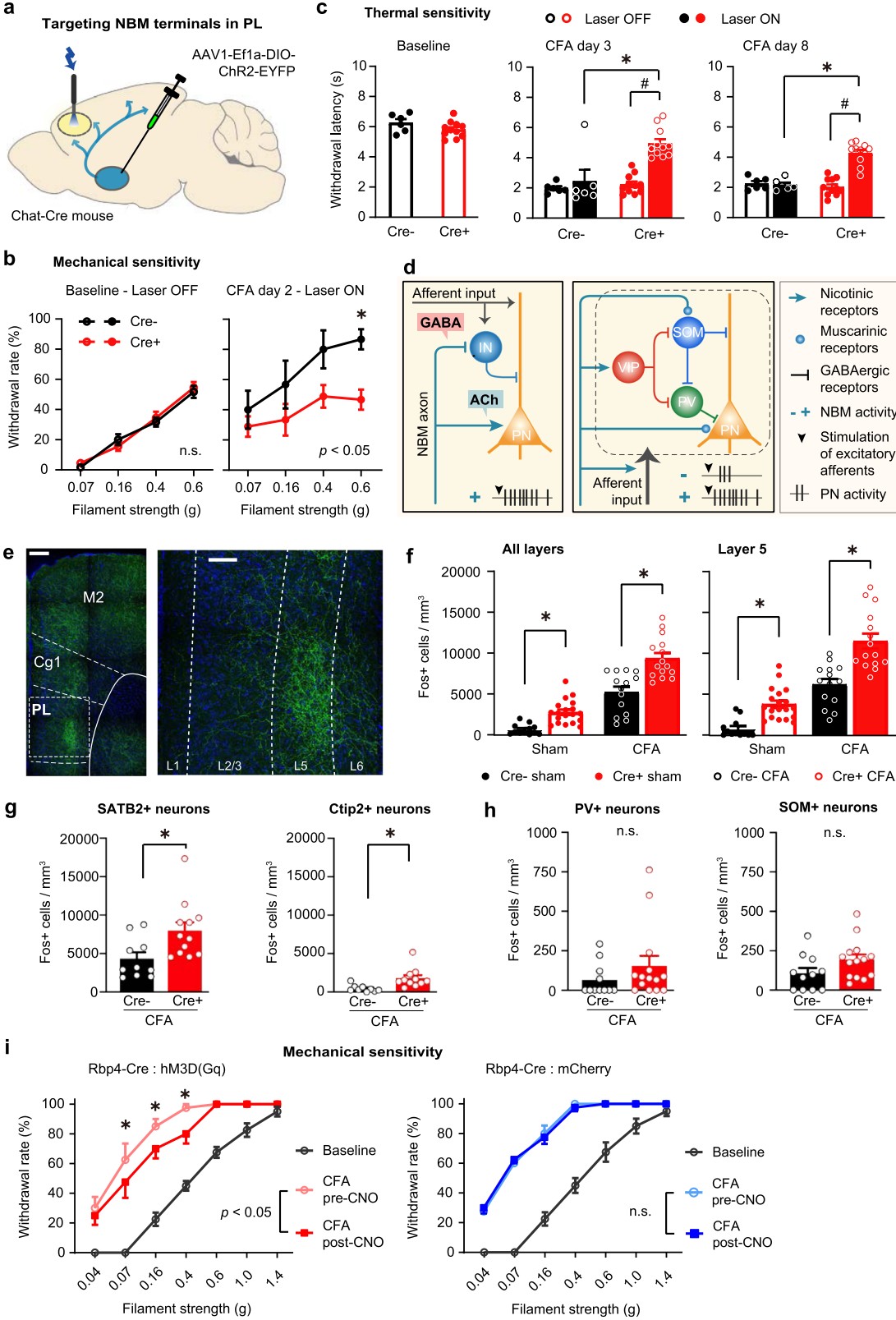

then sought to dissect the significance of NBM neuronal projections to the medial prefrontal cortex (mPFC), which represents a key hub in brain circuits underlying pain. The mPFC undergoes marked plasticity in several human clinical chronic pain conditions and particularly, a major focus has emerged on its deactivation observed in some chronic pain patients[15], a finding that is also reported in animal models of neuropathic pain[16–19]. We therefore

expressed ChR2-YFP in ChAT neurons of the NBM and placed the optic fiber for blue light illumination in the prelimbic cortex (PL), the mouse counterpart of the human mPFC (Fig. 5a). Selectively activating NBM cholinergic projections to the PL did not influence baseline mechanical sensitivity, but had an even stronger analgesic effect than direct activation of NBM neurons on CFA-induced mechanical hypersensitivity, which was completely reversed to

**Fig. 5 | Strong attenuation of inflammatory mechanical and thermal hypersensitivity by optogenetic stimulation of NBM cholinergic-GABAergic projections to the prelimbic cortex (PL) and the role of layer 5 PL neurons.**
**a–c** Scheme for optogenetically stimulating NBM-PL cholinergic-GABAergic projections, and its impact on inflammatory mechanical (day 2; **b**) and heat hypersensitivity (**c**); *P* values in inset (**b**) represent ANOVA-based comparison of the two entire stimulus–response curves. *n* = 6 ChAT-Cre- and 11 ChAT-Cre+ mice; *P* < 0.05 (*0.0043, right panel in **b**; *,# < 0.0001 for CFA days 3 and 8 in (**c**)), two-way ANOVA with Sidak's multiple comparisons test. **d** Scheme of connectivity between NBM cholinergic-GABAergic projections, excitatory afferents and diverse PL neurons impacting on the firing of PL pyramidal neurons via nicotinic and muscarinic cholinergic signaling. IN GABAergic interneuron, PN pyramidal neuron; types of GABAergic interneurons: PV parvalbumin-type, SOM somatostatin-type, VIP vasoactive intestinal peptide-type. **e** Example images of anti-YFP immunohistochemistry showing NBM projections to the PL (magnification to the right) and adjoining cortices (left: different high-resolution confocal fields stitched together). Scale bars = 250 μm (left) and 100 μm (right). **f–h** Fos quantification of all PL layers or layer 5 (**f**) and in excitatory projection neurons (SATB2- or Ctip2; **g**) and inhibitory neurons (PV or SOM; **h**) in response to optogenetic stimulation of NBM-PL cholinergic-GABAergic projections (represented by Cre+ mice) or control (represented by Cre- mice) (all mice are "Laser ON"). *N* = 5 mice/group for panels **f–h**; *P < 0.05 (all layers: 0.0091, Sham, <0.0001, CFA; layer 5: 0.0036, Sham, <0.0001, CFA), two-way ANOVA followed by post hoc Sidak's test for **f**, and unpaired two-tailed *t* test for **g** (*P* = 0.0166, for SATB2 & 0.0066 for Ctip2) and **h**. Only CFA-injected mice are represented in **g**, **h**. **i** Inflammatory mechanical hypersensitivity in Rbp4-Cre mice expressing the excitatory DREADD, hm3D(Gq) in a Cre-dependent manner. *n* = 8 mice/group; *P < 0.05 (0.0019, 0.0019, 0.0002 for 0.07, 0.16, & 0.4 g filaments, respectively), two-way ANOVA with Sidak's multiple comparisons test. Data are presented as mean +/− SEM.

baseline values (Fig. 5b). Furthermore, thermal hypersensitivity was significantly suppressed over a long period post-CFA in mice with optogenetic stimulation of the NBM-PL projections (Fig. 5c).

Electrophysiological and modeling studies indicate that cholinergic inputs from the basal forebrain exert direct excitatory effects via receptor-mediated signaling on cortical pyramidal neurons and also have the ability to evoke either inhibition or disinhibition of pyramidal neurons via signaling on different classes of local neocortical GABAergic interneurons or via signaling through different types of nicotinic and muscarinic receptors[14,20]. Interestingly, recent studies also indicate that GABA is co-released from cholinergic projections originating from the basal forebrain nuclei and can thus disinhibit neocortical pyramidal neurons by suppressing local inhibitory interneurons[14,21] (schematic in Fig. 5d). Both mechanisms have been suggested to act towards enhancing cortical signal-to-noise processing of sensory inputs, e.g., to visual inputs in the visual cortex and tactile inputs in the somatosensory cortex[21]. To address how NBM-PL cholinergic-GABAergic projections affect the PL, we performed viral tracing and c-Fos mapping across layers in conjunction with optogenetics. Interestingly, while NBM projections to most of the neocortical mantle diffusely span all cortical layers, our tracing analyses revealed that projections from the NBM to PL terminate in layer 5 in a particularly abundant manner as compared to other layers (Fig. 5e; compare with neighboring motor cortex M2 and cingulate cortical domains). In both baseline conditions (naive mice) and mice with inflammatory pain-like behavior, optogenetic stimulation of NBM-PL projections led to a significant increase in Fos levels throughout the PL, including layer 2/3, layer 5 and layer 6 (Fig. 5f and Supplementary Fig. 4a; all data points represent 'laser ON' conditions). Fos-expressing PL neurons increased upon paw inflammation as compared to baseline levels in all mice; however, optogenetically activating NBM-PL connections in ChR2-expressing mice enhanced the number of Fos-expressing PL neurons in CFA-injected mice beyond the increase induced by the inflammation (Fig. 5f and Supplementary Fig. 4a; all data points represent "laser ON" conditions). We then performed a detailed characterization of the nature of Fos+ neurons in CFA-injected mice using anti-SATB2 antibody to label excitatory neurons that show a relative preference for association neurons that project intra-telencephalically to other neocortical areas[22] and anti-Ctip2 antibody to label excitatory neurons projecting subcortically[23], and anti-Parvalbumin and anti-Somatostatin antibodies to label the two most abundant populations of GABAergic neurons[11]. We observed that optogenetic stimulation of the NBM-PL projections enhances Fos activation in both SATB2 and Ctip2 populations of excitatory projection neurons (Fig. 5g), but does not alter the activity of GABAergic neurons in the PL (Fig. 5h). Recent studies have shown that a key output of the PL comprises of layer 5 pyramidal neurons in the PL that project to the periaqueductal gray and thereby link to descending nociceptive modulatory systems[24]. We therefore selectively stimulated layer 5 neurons chemogenetically by virally

directing hM3D(Gq) expression[25] in the layer 5-specific Rbp4-Cre line and observed that selectively enhancing layer 5 outputs in the PL mimics the antihyperalgesic action of NBM-PL stimulation on mechanical allodynia in mice with CFA, while baseline sensitivity was not altered (Fig. 5i, left panel; mCherry expression was employed as control, Fig. 5i, right panel).

## Targeting the NBM in a chronic pain model

From the view point of translational relevance, it is important to address whether targeting the NBM cholinergic system is also beneficial in other forms of pain, particularly neuropathic pain. Hence, we addressed whether the cholinergic NBM system is recruited in the neuropathic pain state and is related to mechanical allodynia by studying Fos expression in the absence of or upon plantar application of low-intensity mechanical von Frey force, which is innocuous under baseline conditions. Fos expression in the NBM was significantly elevated in neuropathic mice as compared to sham-injured mice and showed a further increase upon paw stimulation associated with mechanical allodynia (Fig. 6a, b). Importantly, this was also reflected in ChAT-expressing cholinergic neurons (Fig. 6a, b), which suggests increased recruitment by stimuli that are innocuous in baseline conditions but perceived as noxious in neuropathic pain conditions, thus showing parallels to our findings in electrophysiological experiments in the inflammatory pain model described above.

To test the functional significance of these findings in the context of neuropathic pain conditions, we employed a chemogenetic approach to activate cholinergic neurons of the basal forebrain, which provided two advantages: one, it enabled targeting a larger area than optogenetic stimulation (which is limited owing to the maximum area that can be sufficiently illuminated), and second, it permitted achieving more long-lasting activation of cholinergic neurons. ChAT-Cre transgenics were injected with rAAV expressing either the excitatory chemogenetic actuator (mcherry-tagged hM3D(Gq)) or control (mCherry) protein in a Cre-dependent manner and treated with Clozapine N-Oxide, which enables inducible activation of hM3D(Gq)[25] (Fig. 6c). Dual immunohistochemistry for ChAT and Fos demonstrated that 77% of cholinergic neurons in the NBM expressed hM3D(Gq) and 74% demonstrated Fos expression upon CNO treatment, while less than 5% showed Fos expression in the absence of CNO (Fig. 6d), thereby validating the efficacy and specificity of chemogenetic activation of NBM cholinergic neurons. Consistent with data from optogenetics experiments, we observed that baseline nociceptive sensitivity to mechanical pressure and heat was not altered; however, tonic pain-like behavior induced by plantar injection of capsiacin was reduced significantly upon chemogenetic activation of cholinergic neurons (Fig. 6e–g).

We then employed the chronic constriction injury (CCI) model involving unilateral loose ligation of the sciatic nerve,

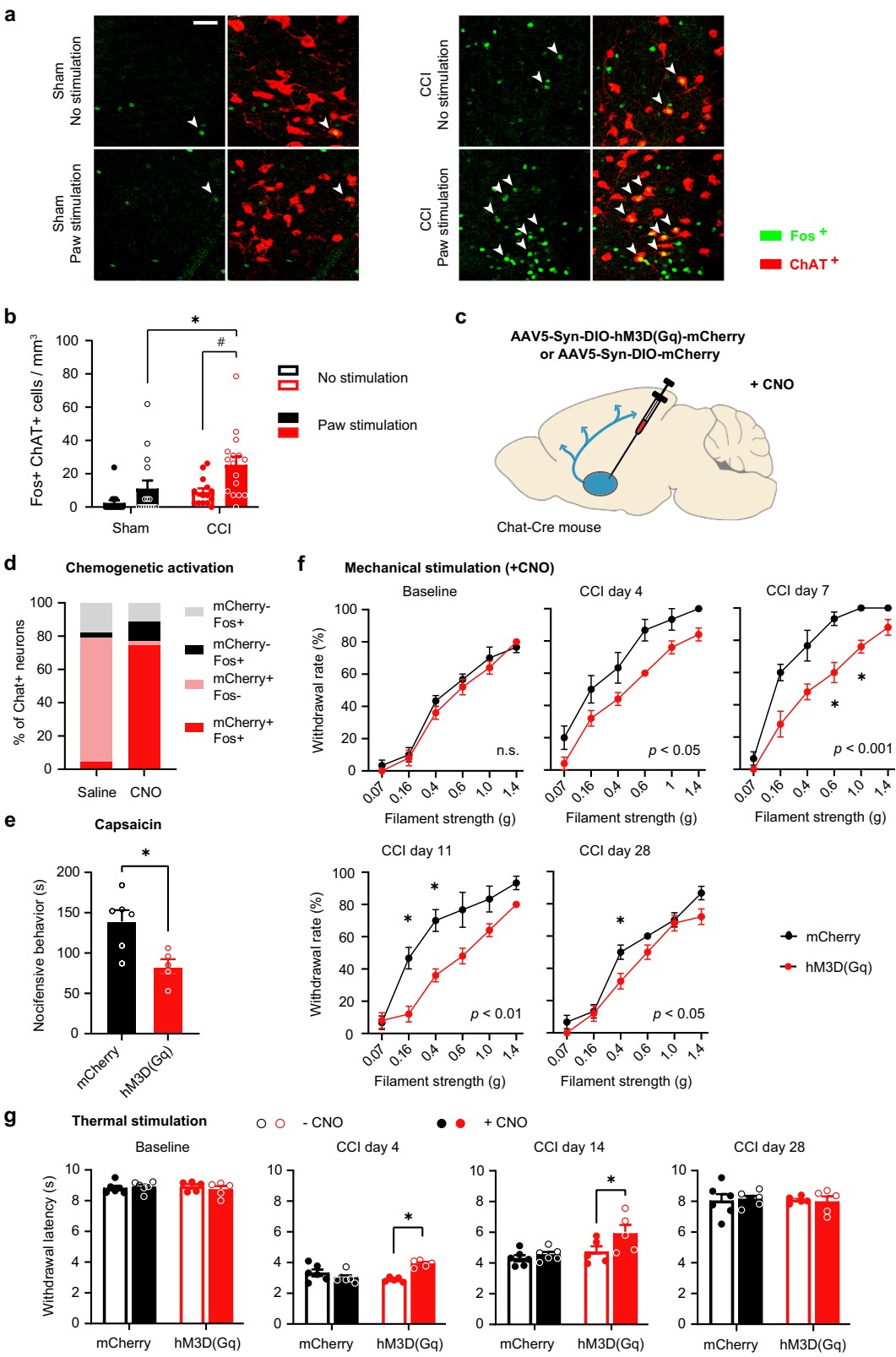

leading to local inflammation and swelling of the nerve, neuropathy and nociceptive hypersensitivity lasting up to a month[26]. CCI-induced mechanical hypersensitivity was markedly reduced in hM3D(Gq)-expressing mice in comparison to mCherry-expressing mice starting from day 4 after CCI surgery (Fig. 6f). At day 11, mechanical responses in hM3D(Gq)-expressing mice were indistinguishable from baseline sensitivity, while mCherry-expressing mice continued to show mechanical hypersensitivity and regained baseline values only on day 28 (Fig. 6f). Similarly, heat hypersensitivity was significantly reduced in hM3D(Gq)-expressing mice as compared to mCherry-expressing mice until day 14 post-CCI (Fig. 6g). These data show that activation of the NBM suppresses neuropathic hypersensitivity over a long duration.

**Fig. 6 | Enhanced recruitment of NBM cholinergic neurons in a neuropathic pain model and attenuation of neuropathic allodynia by chemogenetic activation of the NBM. a, b** Typical examples and quantitative summary of expression of the activity marker Fos in cholinergic neurons (ChAT-expressing) of the NBM in naive mice and mice with chronic constriction nerve injury (CCI) in the presence or absence of a low-intensity (0.16 g) mechanical stimulus at the contralateral hindpaw. Scale bar = 50 μm; $n = 4$ mice/group; $P < 0.05$ (*0.0168, #0.0065), two-way ANOVA with Sidak's multiple comparisons tests. **c, d** Scheme for chemogenetically activating NBM cholinergic neurons expressing hm3D(Gq) with clozapine N-oxide (CNO; **c**) and validation of its efficacy in increasing Fos expression in comparison to mice expressing mCherry in cholinergic neurons (control; **d**). **d** $n = 3$ mice/group. **e–g** Comparison of capsaicin-induced nocifensive responses (**e**) and the development of CCI-induced mechanical hypersensitivity (**f**) or thermal hypersensitivity (**g**) between mice with chemogenetic activation of NBM cholinergic neurons and control (mCherry) mice. $P$ values in inset (**f**) represent ANOVA-based comparison of the two entire stimulus–response curves. $n = 5$ sham & 6 hM3D(Gq) mice; *$P < 0.05$ (0.0121 in **e**; 0.0178 & 0.0231, CCI day 7; 0.0151 & 0.0165, CCI day 11; 0.0209 CCI day 28 in **f**; <0.0001, CCI day 4; 0.010, CCI day 14 in **g**), two-way ANOVA with Sidak's multiple comparisons test. Data are presented as mean +/− SEM.

## Potential contributions of modulation of anxiety, attention, and motor function

Activity of NBM neurons has the potential to affect pain processing directly via cholinergic signaling in neocortical targets that are important in pain networks, as indicated by our observations above on prefrontal layer 5 neurons. However, because the NBM is known to be a key modulator of circuits underlying arousal and attention[20], there is also a possibility that the observed antihyperalgesic effects are related to attention and expectancy. We therefore also addressed whether optogenetic stimulation of NBM cholinergic neurons affects attention behavior using the widely accepted five-choice serial reaction task test (5-CSRT; Fig. 7a) under the same conditions that were implemented in mice used for nociceptive testing in the experiments described in Figs. 4–6. Over 3 weeks, water-restrained mice were trained to learn in operant conditioning tasks to make correct decisions to receive a water reward (Fig. 7a). Chemogenetic stimulation of NBM cholinergic neurons significantly enhanced the accuracy of reactions and reduced the rate of omissions, indicating a higher attention level (Fig. 7b). However, when we optogenetically stimulated NBM-PL projections under the same conditions that were employed in the analyses of pain-like behavior, there was no significant impact on accuracy of reactions and the rate of omissions (Fig. 7c and Supplementary Fig. 5a, b), suggesting that the manifestation of antihyperalgesic behavior was not per se related to attentional alterations. It is, however, plausible that a potential improvement of attention was missed in the experiments related to NBM-PL projections owing to a ceiling effect under baseline conditions, and the situation may be different in a pain state associated with impaired attention. We, therefore, tested mice with inflammatory pain-like behavior in the 5-CSRT test and observed that all mice show a significant increase in omission rate up to 3 days post-CFA as compared to baseline conditions (Fig. 7d). Optogenetically activating NBM-PL projections in these mice, however, did not rescue persistent pain-associated attentional deficits (Fig. 7d and Supplementary Fig. 5c) and CFA-induced impairment was maintained regardless of whether laser stimulation was on (Fig. 7d) or off (Supplementary Fig. 5c) in both ChAT-Cre or Cre-negative control mice, thus further suggesting that NBM-PL projections induce analgesia independently of attentional modulation.

Second, the NBM receives direct inputs from the CeA and is linked to neural circuits of anxiety and fear[27]. We observed that neither direct chemogenetic (Fig. 7e) nor optogenetic stimulation of NBM neurons (Fig. 7f) induced fear-associated behaviors in the open-field test (center-to-margin ratio was unchanged). In contrast, optogenetically stimulating NBM-PL projections reduced the center-to-margin ratio in naive mice, suggesting anxiolytic effects (Fig. 7g). Finally, locomotion was unchanged in all of the groups involving optogenetic or chemogenetic modulation of the NBM or NBM-PL circuits, suggesting a lack of confounding effects on motor function in behavioral analyses (Fig. 7e–g).

## Discussion

Literature on the basal forebrain cholinergic nucleus and pain perception is surprisingly scarce. To date, fewer than a handful of studies have tested the activity of the basal forebrain cholinergic nucleus upon noxious stimulation[28,29]. In this study, we now report the precise nature of oscillatory rhythms in the NBM as well as a detailed analysis at a single-cell level in vivo, showing that the NBM not only responds to noxious stimuli, but also undergoes dynamic changes during the transition to a chronic pain-like state. The most interesting observation was that the power of gamma oscillatory activity in the NBM is specifically enhanced in conjunction with noxious stimulation prior to the behavioral response. Gamma oscillations in the S1 have been functionally linked to nociceptive modulation in both human and rodent systems in vivo[30,31], and these pain-related alterations in gamma activity have only been recently extended to other neocortices, such as the prefrontal and insular cortices[9,32,33]. This study, to the best of our knowledge, represents the first report linking gamma rhythms in a sub-cortical structure to nociceptive sensitivity. These have been likely missed in human studies owing to technical limitations from scalp recordings.

Importantly, our observation that in an inflammatory pain model, the power of gamma activity is potentiated in response to non-noxious tactile stimulation correlates with the manifestation of mechanical allodynia and mimics similar observations made in the S1 cortex[31]. Taken together with current knowledge, a tantalizing implication of our findings is that gamma activity in the NBM is functionally linked via cholinergic pathways to neocortical gamma oscillations during nociceptive processing. This is supported by several conceptual points and experimental observations. First, a recent study in rats reported hemodynamic blood flow changes in the NBM following noxious stimulation and demonstrated that cerebral blood flow changes in the ipsilateral S1 cortex evoked by noxious stimulation were significantly reduced upon lesioning the NBM[34], thus suggesting importance of the NBM in the full manifestation of pain-related responses in the somatosensory cortex. Second, in both S1 and the prefrontal cortex, cholinergic signaling facilitates or even directly elicits gamma-band oscillatory activity via modulation of local GABAergic interneurons[35,36], thereby enhancing the acuity of stimulus processing in sensory and attentional networks, although these phenomena have not been addressed in the context of pain so far. Moreover, gamma oscillatory activity has been proposed to coordinate and link activity states across distant sites in the brain, which is a particularly noteworthy concept in the context of pain[8], since pain is essentially a network function[37,38].

A salient role in the emergence of gamma oscillatory activity is attributed to fast-spiking GABAergic interneurons, which are extensively interconnected via gap junctions; they not only streamline and synchronize excitatory output within a region, but are also capable of doing so at distant sites via long-range GABAergic projections, which typically synapse on GABAergic neurons thereby leading to disinhibition[8,11]. Importantly, here, we observed that while different sets of NBM neurons showed excitation or inhibition upon nociceptive stimulation, the NBM neurons undergoing significant changes during the transition to nociceptive hypersensitivity were derived from waveform analysis to be fast-spiking GABAergic neurons. In the NBM, an overwhelmingly large majority of cholinergic neurons are GABAergic[11] and these comprise long-range projections to the neocortical mantle, thus providing further credence to the association between the origins of gamma oscillatory activity in the NBM and

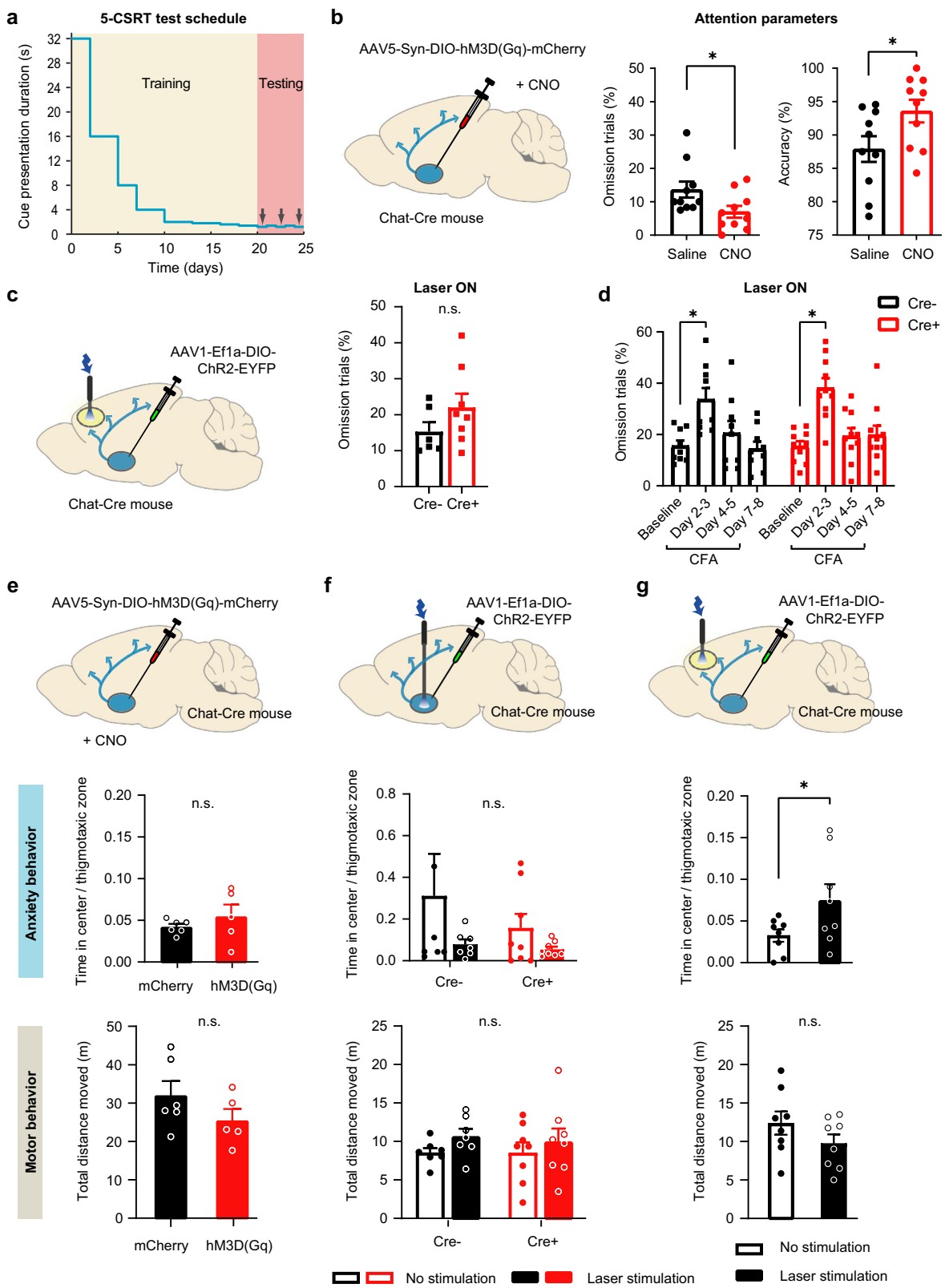

cortical modulation of pain. We also observed changes in the beta frequency range of oscillatory activity; however, much less is known about its cellular origins and functional significance to pain. Studies in healthy subjects have reported that activity in the low-frequency bands, particularly in the alpha and beta ranges, is suppressed in the S1, prefrontal and insular cortices in correlation to subjective pain ratings[9,39]. More work will be needed to unravel the significance of beta

rhythm changes in the NBM in pain states and whether and how this contributes to changes in beta oscillations in the neocortex.

Our analyses with the activity-induced immediate early gene product, Fos, suggest that cholinergic neurons of the NBM are increasingly recruited in inflammatory and neuropathic pain states, particularly in conjunction with sensory stimulation. This finding as well as the overall functional profile of NBM cholinergic neurons

**Fig. 7 | Analysis of attention, anxiety and motor function in mice with diverse manipulations of NBM cholinergic neurons. a** Scheme of training mice in attention-related tasks in the 5-Choice Serial Reaction Task test (5-CSRT). **b, c** Increased attention-related parameters in mice with chemogenetic activation of cholinergic neurons in the NBM (**b**), but not in mice with optogenetic activation of the NBM cholinergic projections to the PL (**c**). $n = 10$ mice/group (**b**); $n = 6$ ChAT-Cre- and 8 ChAT-Cre+ mice (**c**); *$P < 0.05$ (omission trials, 0.0259; accuracy, 0.0008), paired two-tailed $t$ test. **d** Alterations in attentional behavior upon induction of inflammatory pain-like behavior following hindpaw CFA injection as compared to baseline behavior in presence of blue light stimulation of NBM-PL cholinergic projections in ChAT-Cre+ mice and Cre- control mice. $N = 9$ ChAT-Cre- and 10 ChAT-Cre+ mice; *$P < 0.05$ (Cre-, 0.0068; Cre+, <0.0001), two-way ANOVA with Sidak's multiple comparisons test. **e**–**g** Analysis of anxiety-related behavior (upper in **e, f, g**) and locomotion (lower in **e, f, g**) in the open-field test in mice with chemogenetic (**e**) or optogenetic (**f**) activation of cholinergic neurons in the NBM or mice with optogenetic activation of the NBM cholinergic projections to the PL (**g**), as compared to their respective control groups. $n = 6$ mCherry and 8 hm3D(Gq) mice (**e**); $n = 7$ ChAT-Cre- and 8 ChAT-Cre+ mice (**f**); $n = 6$ mice/group (**g**); *$P < 0.05$ (0.0182 for anxiety score in **g**), two-way ANOVA with Sidak's multiple comparisons test. Data are presented as mean +/− SEM.

discussed above could equally well argue for a role for the NBM in either pro-nociceptive or antinociceptive modulation. Here, two independent modes of activation of NBM cholinergic neurons revealed that the net result of their activation is to suppress nociceptive hypersensitivity. A previous study on large-scale damage to cholinergic neurons using conjugated saporin administered via intracerebroventricular injections reported reduced voluntary escape from both noxious heat as well as stressful sound without changes in spinal nocifensive behaviors, concluding that affect, but not sensory behaviors, are modulated by forebrain cholinergic neurons[40]; however, those conclusions were based on widespread ablation across the brain, parenchymal toxicity and loss of connectivity to a large number of areas, including the hippocampus, amygdala and the cortex. Here, cell-specific and reversible manipulations of activity, rather than neuronal integrity, that were limited to the NBM suggest that recruitment of NBM cholinergic neurons subserves an overall protective role in limiting perceived pain. Because NBM neurons project to diverse neocortical domains with differing functions as well as to the amygdala, it cannot be ruled out that individual connections may play different roles. Along these lines, a recent study addressing a different cholinergic nucleus, namely the medial septal nucleus, reported that both inhibition as well as activation paradoxically led to suppression of pain-like behavior by opposing effects on the rostral anterior cingulate cortex and the ventral hippocampal CA1 region[41]. Here, specifically activating NBM projections to the PL cortex made an unequivocal case for an antinociceptive function, which was accompanied by the observations of denser targeting of afferent projections to layer 5 and enhanced Fos expression in layer 5 pyramidal neurons of the PL. As further support, we provide evidence that enhancing output of layer 5 PL neurons via optogenetic stimulation has a similar antinociceptive outcome as stimulation of NBM-PL connections. Our data indicate that the PL shows increased activity in an inflammatory pain-like state, which is consistent with enhanced oscillatory activity in the NBM in inflammatory pain. Importantly, the activity of PL excitatory neurons, but not of inhibitory neurons, is further enhanced by activation of the NBM-PL pathway. This property may hold clinical relevance, since deactivation of the PL has been in reported in some types of chronic pain patients[15,19] as well as rodent neuropathic pain models[16-19]. In neuropathic pain models, enhancing PL output, either optogenetically[24,42], pharmacologically[43] or via non-invasive transcranial brain stimulation[44] has been demonstrated to elicit analgesia. Indeed, the mechanisms underlying deactivation of the prefrontal cortex in chronic pain are a topic of intense current interest, with studies demonstrating enhanced feedforward inhibition via potentiation of amygdalar inputs onto fast-spiking GABAergic neurons in neuropathic pain[17,24]. In contrast, incoming cholinergic afferents from the NBM, which are known to co-release GABA, have the ability to both inhibit or disinhibit neocortical pyramidal neurons via direct modulation vs. modulation of local GABAergic neurons, respectively. Our results now show that cholinergic-GABAergic projections of the NBM to the PL serve to enhance PL activity via increased activation of excitatory projection neurons, which could help to counteract deactivation of the PL in neuropathic pain states. In support, there is evidence from an ex vivo study showing reduction in synaptic expression

of excitatory M1 muscarinic receptors in layer 5 neurons of the PL in neuropathic mice[45] and a study reporting suppression of neuropathic allodynia upon application of a M1–M4 agonist in the anterior cingulate cortex[46]. It will be interesting in future studies to dissect the relative contributions of GABA and acetylcholine, which are co-released from NBM-PL afferents, in modulating the activity of the PL, e.g., via regional delivery of pharmacological agonists and antagonists.

Because the NBM subserves various different functions, it is important to temper our inferences with alternative interpretations. Modulation of attention is one of the best-studied functions of the NBM[20], which was indeed confirmed in this study in mice receiving direct optogenetic stimulation of NBM cholinergic neuronal somata. Attention has been suggested to profoundly modulate pain perception, e.g., by altering descending modulation or in terms of enhancing pain perception by hypervigilanz[47,48], rendering it possible that attentional factors play a role in modulation of pain by the NBM. However, selective activation of NBM cholinergic projections to the PL was insufficient to modulate attention and motivation, and is thus unlikely to fully account for changes in nocifensive behavior. Instead, the enhanced activation of layer 5 PL neurons by cholinergic-GABAergic NBM afferent input would render it more plausible that the known direct connectivity of layer 5 pyramidal neurons to the PAG[18,24] leads to descending modulation of nociception. Motor activity or overall activity were unchanged, suggesting that the observed analgesic effects were independent of arousal and motor dysfunction. The potential anxiolytic effects observed upon stimulating NBM-PL projections are interesting, since they may particularly hold promise in addressing fear as a comorbidity of pathological pain[49].

In total, the results of this study support further investigation of employing cholinergic modulation in pain treatment. While drugs targeting cholinergic receptors have been demonstrated to show efficacy in preclinical models, the large range of side effects impede clinical application. Inhibitors of the acetylcholine esterase, such as Rivastigmine and Neostigmine, which enhance the bioavailability of this neurotransmitter at the sites where it is physiologically released, have shown clinical efficacy in a number of studies in pain[3,50]. It is therefore imperative to delineate the loci of action as well as alterations in the circuitry in chronic pain that show the highest promise in terms of beneficial effects whilst producing fewer side effects. Using cutting-edge in vivo electrophysiology and specific circuit manipulations, this study demonstrates that the NBM is an important modulator of circuitry involved in pain perception and that directly enhancing NBM activity or its projections to the PFC, e.g., via novel designs in neurostimulation techniques, holds promise in the treatment of inflammatory and neuropathic pain disorders. Along these lines, it is noteworthy that the NBM have already been implicated as a potential site of action of general anesthetics[51].

Targeting the NBM may also hold therapeutic promise in the context of other aspects of chronic pain that were not addressed in this study. For example, chronic pain conditions are frequently accompanied by sleep disorders, which are pathogenic in worsening their prognosis and treatment response[52,53]. Sleep deprivation has been shown to lead to reduced connectivity of the NBM to the prefrontal cortex[54]. Furthermore, neuronal loss in the NBM has been reported in

disorders such as Alzheimer's disease and Parkinson's disease, and during aging[55–58]. The results of this study suggest that neuronal depletion in the NBM will likely lead to a loss of central antinociceptive modulatory effects, thereby contributing to pain disorders that are frequently associated with these states. Our observation of reduction in NBM activity over very late stages after paw inflammation are also interesting in this regard. The ongoing development and testing of deep brain stimulation of the NBM[56,58] thus holds promise in not only reducing cognitive decline, but also suppressing pain and restoring normal sleep in these disorders.

## Methods

### Animals

Experiments were performed in male and female 2 to 8-month old heterozygous Chat-IRES-Cre mice (B6;129S6-Chat^tm2(cre)Lowl//Uhg[59];) with a C57BL/6 background, and referred to here as ChAT-Cre mice. Cre-(negative) littermates were used for control experiments. Two- to four-month-old male and female Rbp4-Cre animals (B6.FVB/CD1-Tg(Rbp4cre)^KL100Gsat/Uhg) with a C57BL/6 background were used for chemogenetic targeting layer 5 pyramidal neurons in the prelimbic cortex. C57BL/6J male and female animals aged 8–20 weeks were purchased from Janvier Labs. Animals were housed with food and water ad libitum on a 12 h light/12 h dark cycle. ARRIVE guidelines were followed. All experimental procedures were performed according to the ethical guidelines set by the local governing body (Regierungspräsidium Karlsruhe, Germany; approval numbers 35-9185.81/G44/17 and 35-9185.81/G184/18).

### Surgical procedures

Mice were deeply anaesthetized by intraperitoneal injection of fentanyl (0.01 mg/kg), medetomidine hydrochloride (0.3 mg/kg), midazolam (4 mg/kg). Lidocaine (10%) was applied to the surface of the skin, and a small hole was drilled above the region of interest. In vivo delivery of recombinant adeno-associated virus (rAAVs) was performed by stereotactic injections. The NBM coordinates used relative to bregma were posterior 0.35 mm, lateral 1.6 mm, at a depth of 4.55 mm from the pia. rAAV2-EF1a-DIO-hChR2(H134R)-EYFP (University of North Carolina Vector Core, USA) virus (250 nl) was delivered over 20 min undiluted whereas rAAV5-Syn-DIO-hM3D(Gq)-mCherry and rAAV5-Syn-DIO-mCherry (Addgene Inc., USA) viral solutions were diluted 1:1 in PBS and 400 nl injected over 20 min. Animals were kept for at least 3 weeks to achieve optimal in vivo viral expression prior to behavioral and electrophysiological experiments.

For optogenetic experiments, a chronic optical fiber implant (200 μm core diameter, numerical aperture (NA) of 0.5) was inserted 100 μm above the site of the viral injection in the NBM, or bilaterally in the PL cortex using a lateral rotation of 15° (1.94 mm anterior from bregma, 0.9 mm lateral, at a depth of 1.5 mm from the pia), and secured on the skull with dental cement and a screw. General anesthesia was antagonized with intraperitoneal doses of Naloxone (Inresa Arzneimittel, Freiburg, Germany; 0.4 mg/kg), Flumazenil (Fresenius, Bad Homburg, Germany; 0.5 mg/kg) and Atipamezole (Prodivet pharmaceuticals, Belgium; 2.5 mg/kg).

For electrophysiological experiments, two stainless steel screws were implanted on the skull above the cerebellum and right sensory cortex to serve as ground and reference electrodes, respectively. A cranial window was then prepared above the left basal forebrain (AP = −0.35 mm, ML = 0.9 mm). The dura was removed and versadrive-4 (Neuralynx), composed of four independently drivable tetrodes (Tungsten, 12 μm in diameter, California Fine Wire), was implanted into the left basal forebrain at an initial depth of 4.3 mm. The cranial window was covered with bone wax and the versadrive-4 setup fixed to the skull with dental cement.

For the chronic constriction injury (CCI[26]), mice were placed under isoflurane anesthesia (2%) and the fur of the right thigh was shaved. An incision was made to the lateral skin surface of the thigh and through the biceps femoris muscle to expose the sciatic nerve just above it branches into sural, common peroneal, and tibial nerves. Four loose ligatures were placed around the sciatic nerve using cat gut surgical sutures, the muscle and skin subsequently sutured close, and animals left to recover in a heated cage for 24 h. Behavioral testing commenced from day 2 after the operation. The same surgery was performed without placing the sciatic nerve ligatures on sham control animals.

### Optical stimulation

Mice were restrained securely in a soft cotton cloth in order to attach the optical patch cables (0.5 NA dual fibers connected to a 1 × 2 wavelength division fiber optic rotary joint, Doric Lenses Inc., Canada) to the optical fiber implants. The fiber optic rotary joint in turn was coupled via an optical patch cord (200-μm core diameter, Thorlabs GmbH) to a 473-nm laser (Shanghai Laser & Optics Century Co. Ltd, China). The laser intensity was set to 4 mW as measured at the fiber tip with a light meter (PM100D, Thorlabs). Pulsed laser light (20 Hz, 10-ms pulse duration) was typically applied for a duration of 30 s, starting 15–20 s before mechanical or thermal stimulation events, or with each trial initiation event in the 5-CSRT test, using a pulse generator (cat. no. 33220 A, Meilhaus Electronic GmbH, Germany).

### Behavioral tests

Behavioral tests were carried out during the light cycle of the animals. Animals underwent two acclimatization sessions in the setup chambers used for testing mechanical or thermal sensitivity. Baseline sensitivity was assessed over several days in three test sessions before conducting CCI surgeries or inducing an inflammation of the paw via a subcutaneous plantar injection of 25 μl Complete Freund's Adjuvant (CFA, Sigma-Aldrich) under brief isoflurane anesthesia. Behavioral tests involving animals expressing hM3D(Gq) and corresponding mCherry controls were conducted 1 h after injecting either saline or clozapine N-oxide (CNO, 2 mg/kg intraperitoneal injection; Biomol, Germany). Experimenters were always blinded to the identity of the treatment groups.

**Capsaicin-induced nocifensive behavior.** Capsaicin (Sigma) was diluted from a frozen dimethyl sulfoxide (DMSO, Thermo Fisher Scientific) stock solution (50X) with phosphate-buffered saline (PBS, Thermo Fisher Scientific) to obtain a capsaicin concentration of 0.02% (weight/vol) in 2% DMSO. Animals were briefly anesthetized with 2% isoflurane (Baxter, Germany), and 20 μL of the capsaicin solution was injected with a 30 G needle subcutaneously into the plantar surface of the hindpaw. Animals were then placed in a transparent box (20 × 20 cm) on an acrylic glass plate, and the total time the animal displayed nocifensive behavior (paw lifting, licking, flinching, writhing) was assessed over a period of 5 min by an experimenter blinded to the treatment condition.

**Mechanical sensitivity.** After acclimatization to the von Frey setup (Ugo Basile Inc., Italy), a set of monofilaments that bend at forces of 0.04 g, 0.07 g, 0.16 g, 0.4 g, 0.6 g, 1.0 g, and 1.4 g were applied perpendicular on the plantar surface of the hindpaw. Five applications per filament were applied with a minimum interval of 1 min between each application. In optogenetic stimulation experiments, the laser was turned on 20 s before applying the filament and turned off 10 s after applying the mechanical stimulus. In all, five applications per filament and per laser state were applied during a test session for each animal. Trials were scored positive if the animal exhibited nocifensive response behaviors, including rapid paw withdrawal, licking, or shaking of the paw, either during the mechanical stimulation or immediately after the filament was removed. The paw-withdrawal threshold was determined using Dixon's up-down method[60].

**Thermal sensitivity.** The Hargreaves plantar test setup (Ugo Basile Inc., Italy) with an infra-red heat source (Model 37370-001, Ugo Basile) was used to test thermal withdrawal thresholds by applying radiant heat to the plantar surface of the hindpaw. The intensity level was set to 25 and the cutoff time to 30 s. A heat stimulus was applied only during the quite wake phase and the withdrawal latency from stimulus onset was recorded. Six trials were performed per treatment condition and animal on a test day, using a minimum inter-trial interval of 2 min. Laser ON trials were randomly interspersed with laser OFF trials during an optogenetic Hargreaves test session. The laser was turned on 20 s before initiating the thermal stimulus and switched off 5 s after a paw-withdrawal response.

**Open-field test.** The open-field test was performed in a square box (40 × 40 cm, 38 cm in height) with a USB camera fixed above the box in order to track animal movement patterns and record experimental parameters using ANY-maze software (Stoelting Co., Ireland). The animals were not acclimatized to this setup so that they encountered a novel arena to explore. The box was divided into three zones for analysis. A 3-cm-wide border along the walls of the box was defined as the thigmotaxic zone. A square zone of 20 × 20 cm in the geometric center of the box was defined as the center zone. The remaining area was defined as the marginal zone. Each mouse was placed in the center of the box and allowed to explore the entire field freely for an 8 min period. The 8 min test was divided into 30 s periods with the laser turned ON and OFF alternately in a random tact so that each mouse received 4 min illumination overall. During the test, the locomotion parameters (distance and mean speed) within each segment were recorded. The ratio of the time spent in the center versus the thigmotaxic zones was used to assess anxiety-like behavior. Motor function was assessed from the total distance moved in all three zones.

**5-Choice serial reaction time (CSRT) test.** Three cohorts of ChAT-Cre mice either expressing the hM3(Gq) DRADD ($n = 10$) or the ChR2(H134R) opsin ($n = 8$ and $n = 10$) in cholinergic NBM neurons as well as six and nine Cre-ve control animals for the optogenetic test groups were trained in the 5-choice serial reaction time (5-CSRT) task using automated Bussey-Saksida Mouse Touch Screen operant chambers (Campden Instruments, Loughborough, UK) and ABET II TOUCH software (Lafayette Instrument, IN, USA). Throughout training and testing stages, animals had limited access to drinking water (30 min per day) and hence water could be used as a reward to reinforce correct choice behavior in individual trials during the task. For habituation and training, procedures outlined in Humby[61] and the ABET II TOUCH 5-CSRT task module (version 3) were followed. Briefly, a light cue was presented in one of five windows for a given time period and a trial was counted as correct if the animal touched the monitor of the cued window within an extended time of 5 s after the cue disappeared. If the mouse interacted with another window (incorrect trial) or no touch screen interaction was detected (omission trial) following cue presentation a punishing time out period was signaled by the house light turning on for 5 s. A session consisted of 60 trials and mice performed one session per day. The cue duration was successively reduced from 30 s to 1.4 s until performance at each stage reached a criterion of >80% accuracy [number of correct trials/total number responded trials (correct + incorrect)] and <20% omissions [number of missed trials/number of trials presented] for two consecutive days. DREADD animals were tested three times over a 5-day period using a cue duration of 1.2 s after having received a saline or CNO injection. The cue presentation period was increased by 0.2 s in maintenance sessions between test days.

Optical patch cords were connected daily already during the later training phase of the optogenetic cohort without turning the laser on. Upon reaching the performance criterion with cues displayed for 1.8 s, animals were tested using a cue presentation period of 1.6 s with the laser turned ON at the start of each trial and OFF after collection of the water reward, or immediately after the 5 s extended time window if no correct touch response was detected. After establishing a baseline by testing with or without the laser being turned on, a cohort of ChAT-Cre+ ($n = 10$) and Cre− ($n = 9$) mice received a subcutaneous plantar injection of 20 µl CFA (see behavioral test section above), and were then tested on alternate days with the laser turned on or off, in a semi-random order. Animals that had more than 30% omission trials during the baseline period were excluded.

## Histology and immunohistochemistry

At the end of the experiment, mice were killed with an overdose of carbon dioxide and transcardially perfused with phosphate-buffered saline (PBS) followed by 10% formalin (Merck, Germany). Brains were collected and post-fixed additionally for 24 h at 4 °C. Brain sections were cut with a vibratome at 50 µm thickness, mounted with Mowiol, and imaged with a fluorescent microscope to confirm the location of the electrode sites or the AAV injection in Cre+ animals.

To assess cholinergic neuron activity changes in acute and inflammatory pain-like conditions, animals were perfused 90 min following a capsaicin injection into the hindpaw, repetitive mechanical stimulation (0.16 g filament, 20 s interval over a 10-min period) of the inflamed paw on CFA day 2, and on day 4 to the ipsilateral paw of CCI and sham animals. To assess neuronal activity induced by the optogenetic stimulation, the pulsed Laser light was turned ON five times for 30 s during a 10 min period in Cre+ and Cre- animals that were then perfused 90 min later. Similarly, CNO or saline was injected in DREADD animals 2 h before perfusion.

Dual immunolabelling with anti-Fos (rabbit; ab190289, Abcam, UK) and anti-ChAT (goat; AB144P, Merck) were used at 1:1000 and 1:250, respectively. Briefly, sections were incubated in PBS/50 mM glycine for 10 min, followed by a blocking step of 60 min in 4% horse serum with 0.2% Triton in PBS. Sections were incubated with both primary antibodies in the blocking solution for 24 h at 4 °C. The sections were subsequently washed in blocking solution (three 10 min washes) and incubated with a secondary antibody mixture of donkey anti-rabbit-Alexa-488 and donkey anti-goat-Alexa-633 (Invitrogen, USA; 1:700 each) in blocking solution for 2 h at room temperature. Donkey anti-rabbit-Alexa-594 was used for brains with AAV-induced EYFP expression. To enhance EYFP fluorescence of AAV-transduced NBM terminals a subset of frontal brain sections was immunolabelled with a cocktail of rabbit anti-Fos (1:1000) and chicken anti-GFP (ab13970, Abcam; 1:1000, pre-incubated first for 72 h at 4 °C at a 1:100 dilution with brain sections from naive C57bl6 mice) primary antibodies (48 h incubation at 4 °C), using donkey anti-rabbit-Alexa-594 and goat anti-chicken-Alexa-488 (Invitrogen, USA; 1:700 each) secondary antibodies (as above, incubated for 2 h at room temperature). Tissues were washed in PBS twice, incubated in Hoechst 33342 (diluted 1:10,000 in PBS from 10 mg/ml stock solution, Invitrogen) for 10 min, washed again in PBS, and further incubated for 10 min in 10 mM TRIS-HCl before mounting.

In addition, two sets of triple immunohistochemical labeling experiments were performed with rabbit anti-Fos (Abcam, 1:1000), rat anti-somatostatin (EMD Millipore, 1:300), and guinea pig anti-parvalbumin (Swant, 1:500), or rabbit anti-Fos (Abcam, 1:1000), rat anti-Ctip2 (Abcam, 1:500), and guinea pig anti-SATB2 (Synaptic Systems, 1:200). Sections were processed as described above for the dual immunostaining procedure and incubated for 48 h at 4 °C. Secondary antibodies used were donkey anti-rabbit IgG Alexa-488, donkey anti-rat IgG Alexa-594, and goat anti-guinea pig IgG Alexa 647 (all Invitrogen, 1:700). Specificity of antibody staining was tested by omitting the primary antibodies.

## Imaging and counting

Sections were imaged using a laser-scanning confocal microscope (Leica TCS SP8, Germany) with a pixel resolution of 1024 × 1024. The illumination parameters were kept identically for an image series across all animals. A dry air objective (Leica, 10×/0.40, HC PL APO) was used for imaging Fos-labeled sections and an immersion objective with correction collar (Leica, 20×/0.75, HC PL APO) for imaging triple-labeled sections. A montage of confocal image stacks was acquired over a depth of 25 µm centered over the region of interest at the mid-level of each section and the maximum z-projection of images were applied for counting in ImageJ software (version 1.50b, National Institutes of Health, USA). The mouse brain stereotaxic atlas[62] and the reference atlas from the Allan Institut (2011) were respectively used to define region of interest and cortical layer outlines according to corresponding reference sections. Fos-labeled, double and triple-labeled cells within each region of interest were counted manually using the same contrast and threshold settings for all sections within an experimental group. Positive cells lying on the boundary were excluded. Cell counts were converted to indicate the number of positive cells in the imaged stack volume within the region of interest (cells/mm³).

## Cell counting in the basal forebrain.

A region of interest (1.5 mm mediolateral ×1.0 mm dorsoventral in size) was used as a counting frame above the NBM region with the highest expression of ChAT+ or EYFP+neurons. The lower edge of the globus pallidus was used as an upper boundary. Fos+ double and triple-labeled cells within the counting frame in both hemispheres were counted manually using the same contrast and threshold settings for all sections within an experimental group. Counts of double-labeled Fos+ neurons of the capsaicin treatment group are averages of three brain sections expressed as% of ChAT+ neurons in each hemisphere. As the number of double- and triple-labeled Fos+ neurons did not differ significantly between hemispheres for all other treatment groups, data from the two hemispheres were averaged for each brain slice, and converted to indicate the number of positive cells in the imaged stack volume within the region of interest (cells/mm³).

## Electrophysiology

After one week of recovery from the implantation surgery, tetrodes were lowered down by 0.5 mm on average into the region of interest and remained unchanged until the end of the experiment. Mice were allowed 2 days to habituate to the elevated grid of the von Frey test recording setup. Naive mechanical sensitivity tests were performed with weak (0.07 g and 0.6 g) and strong (0.6 g and 1.0 g) filaments for 4 days. Each filament was applied ten times on the planter surface of the right hindpaw with a minimal 60 s interval between stimulation trials. Chronic inflammation was induced by injecting CFA solution (25 µl, Complete Freund's Adjuvant, Sigma) subcutaneously on the plantar side of the right hindpaw. Mechanical nociception tests were performed on days 1, 2, 3, 4, 7, 9, 12, 14 after CFA injection with the same filaments used for the baseline tests. At the end of the behavioral experiments, mice were deeply anesthetized with 2% isoflurane, the location of each tetrode tip labeled by applying electrical current to induce a small lesion, and the animals perfused transcardially to fix the brain tissue.

Neural signals were acquired via a HS-18-MM headstage using Digital Lynx 4SX system and cheetah data acquisition software (Neuralynx). The raw data was acquired at 32 kHz with a bandpass filer (1–6000 Hz). The von Frey stimulation was recorded by a custom-made piezo transducer (Piezo ceramic element, part #717770, Conrad), which transduced the pressure of von Frey stimulation into an analog signal bandpass filtered at 1–2000 Hz[31]. In addition, videos of mechanical stimulation events were recorded by a USB camera (20 frames/second), synchronized via a keyboard-generated event signal to the piezo signal. Stimulation onset was defined as the time of contact of the von Frey filament with the hindpaw corresponding with an initial deflection of the piezo signal by visually inspecting the video and piezo recordings, respectively.

## Analysis of electrophysiology data.

Local field potential (LFP) and single unit activity were analyzed with custom-written scripts using MATLAB[63] (The Mathworks Inc, Version R2014a). Statistical analysis and post hoc tests were performed in Graphpad Prism (version 9).

## Power spectrogram analysis.

For the spectrogram analysis *of the LFP activity*, 3 s before and 3 s after the onset of the von Frey filament application for withdrawal trials was extracted from the raw data. One channel of a tetrode was analyzed per animal. Raw data episodes were filtered with a 3rd-order lowpass Chebyshev type I filter with 0.5 dB ripples in the passband and a passband edge frequency of 200 Hz and down sampled to 1000 Hz. Power spectrograms were generated with the Morlet wavelets function, setting the central frequency to 0.8125 Hz, frequency accuracy at 0.5 Hz, and the time resolution to 1 ms. The 1 s baseline period before the stimulation onset was used to normalize each 0.5 Hz frequency segment by the respective mean, and expressed as% deviations from the pre-stimulation baseline. The normalized power spectrograms of individual trials were then averaged for weak and strong filaments for each mouse and day. Grand mean averages of these normalized spectrograms for all animals are shown in Figs. 1e, 2c, and Supplementary Fig. 3a.

For the quantitative analysis, averages of four frequency bands, including theta (4–8 Hz), alpha (8–14 Hz), beta (14–30 Hz), and gamma (30–100 Hz), in the power spectrograms of each animal were calculated over the entire 2 s post-stimulation period. For the time-course analysis, normalized spectrograms of each animal were binned into 100-ms bins, and averages calculated for each frequency band. The median withdrawal time of the corresponding trials for a spectrogram was calculated as a reference. To correlate individual filament force with the increase in the spectrogram power, the normalized power over the 2 s post-stimulation was averaged for each filament and frequency band for each animal.

## Single unit analysis.

Spike sorting was performed with Kilosort2[64] to isolate single units. Raw data was pre-processed with a bandpass filter from 300 to 6000 Hz. Drift correction, unit clustering, and template matching was automatically performed based on the template matching method. Automatically clustered units were manually curated in Phy (version 2.0; https://github.com/cortex-lab/phy) using waveform similarity and cluster features, firing rate, as well as cross-correlation and auto-correlation features.

## Detecting stimulation-responsive units.

For the analysis of evoked activity changes in the single unit data, firing activity of each withdrawal trial was aligned to the stimulation onset for the withdrawal trials of either all filaments, or separately for weak and strong filament groups. The firing rate across trials was calculated for 250 ms bins and z-scores computed based on the mean and standard deviation of the 3 s pre-stimulation baseline activities[44]. Units showing significantly increased or decreased activity were identified if at least one of the normalized bins in the 3 s post-stimulation period exceeded 3.09 or −3.09, respectively, corresponding to a significance level of $P < 0.001$. Otherwise, the unit was classified as an unresponsive unit. Units were excluded from this analysis if the mean firing rate was smaller than 1 Hz or the number of withdrawal trials less than 3. In order to compare the magnitude of stimulation-evoked responses, the maximal and minimal z-score was extracted for all units with significantly increased or decreased firing rates within the 3 s post-stimulation periods, respectively.

**Unit type classification.** For the single unit classification, various parameters were calculated, including: firing rate, coefficient of variation of the inter-spike intervals, peak–peak amplitude of the waveform, time between early and late waveform peaks, time from waveform trough to the return to baseline, and waveform asymmetry (the quotient of the difference between the baseline to early peak, and late peak to return of baseline times, to the sum of these two times). These multi-dimensional parameters were projected into two dimensions using the t-SNE (t-distributed stochastic neighbor embedding) Matlab function[65]. Then k-means algorithm was applied to cluster these units into two clusters. Based on cluster separation, the best unit classification was achieved using just two waveform parameters: the asymmetry parameter and the time from trough to the return to baseline. We did not attempt to distinguish if the units we classified were excitatory or inhibitory neurons, nor can we distinguish between projection neurons and interneurons.

### Statistical analysis

All data are expressed as mean ± SEM. unless stated otherwise. Prism (version 9) was used for the statistical analysis of all behavioral data and for performing post hoc comparison tests of electrophysiological data sets. A one-sample t test was performed to detect if specific frequency bands of the LFP power spectrogram of withdrawal trials deviated significantly from the pre-stimulation baseline. The ROUT method with $Q = 0.5\%$ was used to test for outliers. A repeated measures one-way ANOVA with Dunnett's multi-comparison test for differences to the pre-stimulation baseline was used for the time-course analysis in Fig. 1g and Supplementary Fig. 1b. All grouped data sets were analyzed with a two-way ANOVA using Sidak's test for multiple comparisons for relevant treatment combinations that had significant main group effects. The unpaired Student's t test was used to test for treatment effects compared to a control group. The Chi-square contingency test for the unit response types (Supplementary Fig. 2b) was applied for all time periods, as well as for pair-wise time period combinations to detect the deviating data set. In all tests, a P value of <0.05 was considered significant.

### Reporting summary

Further information on research design is available in the Nature Research Reporting Summary linked to this article.

## Data availability

The electrophysiology data generated in this study have been deposited in the heiDATA repository under accession code [https://doi.org/10.11588/data/ET9G9X]. Source data are provided with this paper.

## Code availability

Matlab scripts for analyzing electrophysiological data are available together with the raw data in the same repository [https://doi.org/10.11588/data/ET9G9X].

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

## Acknowledgements

The authors are grateful to C. Gartner for secretarial assistance and to N. Gehrig, V. Buchert, D. Baumgartl-Ahlert, and B. Zimmermann for excellent technical assistance. This work was supported by the Deutsche Forschungsgemeinschaft to R.K. in form of grants in the CRC1158 chronic pain consortium (Project B01, B06). M.O. received a seed grant from CRC1158 funds. The authors gratefully acknowledge scholarship support for Z.G. from Union Hospital, Tongji Medical college, Huazhong University of Science and Technology, for H.L. from China Scholarship Council and P.V.N. from the Studienstiftung des Deutschen Volkes (German National Merit Program) as well as a young scientist career fellowship from CRC1158 funds. The authors gratefully acknowledge the Interdisciplinary Neurobehavioral Core Facility of Medical Faculty Heidelberg for

assistance with behavioral experiments and the data storage service (SDS@hd) supported by the Ministry of Science, Research and the Arts Baden-Württemberg (MWK) and the German Research Foundation (DFG) through grant INST 35/1314-1 FUGG and INST 35/1503-1 FUGG.

## Author contributions

M.O., Y.H., Z.G., H.L., D.U., B.O., S.M., and P.V.N. performed all wet experiments under supervision from R.K. M.O. assisted with the planning and execution of electrophysiology protocols. R.K. conceptualized the project, and all authors provided regular conceptual inputs. M.O., H.L., and Z.G. prepared the figures. R.K. wrote the manuscript and all authors provided comments and methodical information in writing the manuscript and presenting the data.

## Funding

## Competing interests

The authors declare no competing interests.
