## [Peer Review File · Nature Communications]

Cholinergic basal forebrain nucleus of Meynert regulates chronic pain-like behavior via modulation of the prelimbic cortexREVIEWER COMMENTS

Reviewer #1 (Remarks to the Author):

The authors demonstrate that cholinergic projections from the basal nucleus of Meynert (NBM) to the prefrontal cortex play an important role in top-down control of pain processing in models of inflammatory and neuropathic pain.

First, the authors show that GABAergic/Cholinergic NBM neurons are activated (express cFos) in response to inflammatory pain; accordingly, in-vivo electrophysiological recordings from the NBM show increased oscillations in response to nociceptive stimuli.

Then, the authors take advantage of transgenic mice to express channelrhodopsin selectively in cholinergic NBM neurons and show that blue light effectively activates these neurons (induces cFos). They further show that optogenetic activation of NBM neurons induces mechanical analgesia in the early phase (two/three days) after onset of inflammatory pain. This effect is mimicked by activation of NBM terminals in the PFC. The authors further show that chemogenetic activation of the NBM provides robust mechanical and thermal analgesia in a model of neuropathic pain at numerous time points (day 4 to day 14). Finally, they show that in naïve animals, activation of the NBM inputs to the PLC does not have overt effects on attention and anxiety.

Overall, the results convincingly show that cholinergic modulation of the prefrontal cortex by NBM neurons provides a potent modulation of pain perception and may represent an interesting pharmacological target. The rationale for the study is presented well and the results are generally described clearly, although some figures could be clearer. The data appear robust, and the findings are very interesting. There are however a few points that in my opinion deserve further attention by the authors.

- 1- The authors report that chemogenetic stimulation of NBM cholinergic neurons significantly increased attention level in naïve animals; however, this effect was not reproduced by selective stimulation of the NBM-PL projections and they conclude that the analgesic effect was not per se related to attentional alterations. This however is not necessarily the case because it was previously shown that attention is impaired in neuropathic pain animals (Pais-Vieira et al. 2009). Thus, it is possible that activation of NBM-PL projections in control animals has no effect on attention because of a ceiling effect; yet, activation of these projections, which has no effect on attention in naïve mice, may become important for attention in pain conditions, and this may influence analgesia. It would therefore be important to know whether activation of these projections affects attention in CCI mice; alternatively, the authors should at least provide a through discussion of this point.
- 2- The authors show that activation of NBM projections to the PLC induces cFOS expression in layer 5. Is this expression in pyramidal cells, interneurons or both? High magnification images may help.
- 3- I am not sure what the red and black bars represent in Fig. 5 F,G. Is it laser on/off? If so, why does the laser activate cFos expression in Cre-negative animals? Also, if these are pyramidal neurons, why does the inhibition of the PL in pain conditions (see papers from the Neugebauer and Zamponi groups) increase cFos (see sham vs CFA data)? If these are mostly interneurons, then why does their activation cause analgesia?

Reviewer #2 (Remarks to the Author):

Summary: Oswald et al. use a multifaceted approach including optogenetics, in-vivo electrophysiology, confocal imaging, and behavioral tasks to study the role of cholinergic-GABAergic neurons within the basal nucleus of Meynert (NBM) in pain regulation. The authors present new data indicating that these NBM neurons respond to noxious stimuli with activity being enhanced by inflammatory pain. Data is presented indicating that NBM neurons play a role in attenuating pain hypersensitivity via projections to the prelimbic (PL) cortex. The authors also present data suggesting that chemogenetic activation of NBM cholinergic neurons can improve accuracy in operant conditioning tasks.

Originality and significance: Oswald et al. presents intriguing findings that address a significant knowledge gap in how a subset of forebrain cholinergic neurons play a role in pain perception and ongoing pain. While significant, concerns regarding data interpretation and approach limit enthusiasm.

Data, methodology, & statistics: The approach is valid. There are some concerns however.

1. Figure 1d does not show ChAT staining as stated in the text (Page 5, line 4 of Results). I believe this is supposed to be Figure 1c.
2. In Figure 1F, there seems to be a single animal that showed a robust increase in theta, alpha, and beta oscillations. It is surprising that statistical significance was not reached. F and p values should be reported for these data (this was confirmed as being reported in the reporting summary associated with the manuscript). Gamma showed the smallest change, yet it was the focus of Figure 1g.
3. It is confusing how data from Supplemental Figure 1b represents a longitudinal study design. It is showing increased stimuli force.
4. It is not clear what the authors are defining as peak mechanical hypersensitivity when referencing Figure 2c and d. Withdrawal rate % is highest at 0.6-1.0 grams of force, which is not where the largest increase in beta and gamma frequencies are observed (which is extremely variable). It seems as if the authors are defining peak mechanical hypersensitivity as the force at which there is the biggest difference in withdrawal rate between naïve and CFA animals in Supplemental Figure 1b, but this is not at all clear.
5. The mean differences between CFA and Naïve for both classes of neurons in Figure 3d look similar. It would be informative if full statistic values were given as it is possible that the Class 1 analysis is underpowered.
6. It is a bit concerning that the overlap of ChAT staining and EYFP expression in Figure 4C looks relatively low.
7. The explanation of Figure 5f and 5G is lacking. The assumption is that red is laser on and black is laser off. Therefore, in the Cre- animals, there should be no expression of opsin, so it is unclear why there is an increase in Fos+ neurons. Further, it would be informative to report Fos+ neurons in L2/3 and L6 for comparison. Two-way ANOVA with Sidak's multiple comparisons tests should also be used for these comparisons.
8. The organization of Figure 6 is confusing. It looks as if CCI days 7, 11, and 28 are within 6e. Makes it difficult to follow.
9. The use of robustly in the following statement "heat hypersensitivity was robustly reduced in hM3D(Gq)-expressing mice as compared to mCherry-expressing" is subjective. At what point is the change not robust? Should be replaced with the word significantly.

Conclusions: The robustness and validity of the study is considered adequate. Even though it is discussed briefly, there still remains some questions regarding the potential dichotomy of GABA vs. ACh effects in mPFC via release from NBM axons. Use of cholinergic receptor antagonists during experiments may have provided insight into whether behavioral changes were driven by GABA or ACh, or whether aspects of specific behaviors were controlled by GABA and ACh separately.

Clarity and context: Relatively well-written. Use of subjective words like "robustly" could be reduced.

Responses to reviewers by Oswald et al.

The authors are very grateful to all reviewers for the highly constructive, knowledgeable and insightful reviews. The reviewer comments and suggestions were immensely helpful in improving the quality of the reporting and in designing new experiments to address key open questions. We have addressed all concerns and critical points and are delighted to present additional data from new experiments. The manuscript has been stringently revised and we are pleased to report that all new data support the main inferences of the study. Point-by-point responses are given below and the changes in main text are marked in blue-coloured font.

REVIEWER COMMENTS

Reviewer #1 (Remarks to the Author):

The authors demonstrate that cholinergic projections from the basal nucleus of Meynert (NBM) to the prefrontal cortex play an important role in top-down control of pain processing in models of inflammatory and neuropathic pain. First, the authors show that GABAergic/Cholinergic NBM neurons are activated (express cFos) in response to inflammatory pain; accordingly, in-vivo electrophysiological recordings from the NBM show increased oscillations in response to nociceptive stimuli. Then, the authors take advantage of transgenic mice to express channelrhodopsin selectively in cholinergic NBM neurons and show that blue light effectively activates these neurons (induces cFos). They further show that optogenetic activation of NBM neurons induces mechanical analgesia in the early phase (two/three days) after onset of inflammatory pain. This effect is mimicked by activation of NBM terminals in the PFC. The authors further show that chemogenetic activation of the NBM provides robust mechanical and thermal analgesia in a model of neuropathic pain at numerous time points (day 4 to day 14). Finally, they show that in naïve animals, activation of the NBM inputs to the PLC does not have overt effects on attention and anxiety.

Overall, the results convincingly show that cholinergic modulation of the prefrontal cortex by NBM neurons provides a potent modulation of pain perception and may represent an interesting pharmacological target. The rationale for the study is presented well and the results are generally described clearly, although some figures could be clearer. The data appear robust,

and the findings are very interesting. There are however a few points that in my opinion deserve further attention by the authors.

Response: We thank the reviewer for the scholarly and in-depth evaluation and are very pleased to note the overall positive nature of the assessment. Below, we have addressed all of the points the reviewer raised and show additional data. We have also striven to improve the clarity of the figures.

1- The authors report that chemogenetic stimulation of NBM cholinergic neurons significantly increased attention level in naïve animals; however, this effect was not reproduced by selective stimulation of the NBM-PL projections and they conclude that the analgesic effect was not per se related to attentional alterations. This however is not necessarily the case because it was previously shown that attention is impaired in neuropathic pain animals (Pais-Vieira et al. 2009). Thus, it is possible that activation of NBM-PL projections in control animals has no effect on attention because of a ceiling effect; yet, activation of these projections, which has no effect on attention in naïve mice, may become important for attention in pain conditions, and this may influence analgesia. It would therefore be important to know whether activation of these projections affects attention in CCI mice; alternatively, the authors should at least provide a thorough discussion of this point.

Response: The reviewer has indeed raised an excellent point. It is completely plausible that a potential effect on attentional parameters was missed in the experiments related to NBM-PL projections owing to a ceiling effect in control mice, and the situation may be different in a pain state associated with impaired attention. We have followed up on the reviewer's suggestion and show experimental data that tests this hypothesis. In place of CCI, we employed the CFA (Complete Freund's adjuvant) model, since our recent unpublished data showed that attention is also impaired in mice with CFA in a manner similar to that previously described for neuropathic mice and because we had ethical permission to employ this model for attention-related experiments. We clarified this point with the Editor and were given positive feedback regarding using the CFA model.

In revised Fig. 7d, we show data representing an increase in omission rate in the 5-choice serial reaction time test (5-CSRT) in mice with CFA in comparison with control mice. Mice expressed YFP-tagged Channelrhodopsin (ChR2) in ChAT-expressing neurons of the NBM and an optic fiber delivering blue light was placed in the prelimbic cortex (PL). Optogenetically activating cholinergic NBM-PL projections did not significantly rescue attentional deficits in

mice with persistent pain and the parameters were not different from mice expressing mCherry as control in place of ChR2 (Fig. 7d; Suppl. Fig 5c). These results thus further support the notion that inhibition of nociceptive hypersensitivity by activation of NBM-PL projections is not related to alteration of attentional parameters.

These data are shown in revised Fig. 7d and Suppl. Fig. 5c and described on page 9 of the revised manuscript.

2- The authors show that activation of NBM projections to the PLC induces cFOS expression in layer 5. Is this expression in pyramidal cells, interneurons or both? High magnification images may help.

Response: We have now performed detailed analyses of the nature of PL neurons that show increased c-Fos expression upon optogenetic activation of NBM-PL projections. Because Fos expression is limited to the neuronal nucleus, higher magnification images do not reveal the excitatory (projection neuron) nature of the cells. Therefore, we have now immunostained pyramidal neurons in the PL for characteristic marker proteins, namely SATB2, which labels neocortical excitatory association neurons that project intra-telencephalically to other neocortical areas (Alcamo et al. 2008) and Ctip2, which labels neocortical excitatory neurons that project sub-cortically to deeper structure (Molyneaux et al. 2007). The two most abundant classes of GABAergic inhibitory interneurons were identified based upon expression of Parvalbumin and Somatostatin neurons. Typical examples of staining are shown in Suppl. Fig. 4b, c and quantitative overview is given in Fig. 5g, h.

Our results indicate that activation of NBM-PL projections leads to increased activity of neocortical excitatory projection neurons, but not of GABAergic inhibitory neurons. Thus, the NBM-PL pathway differs from the amygdala-PL pathway that recruits GABAergic neurons in the PL and leads to prefrontal deactivation (Huang et al. 2019).

These data are shown in revised Fig. 5g, h and Suppl. Fig. 4 and described on page 10 of the revised manuscript.

3- I am not sure what the red and black bars represent in Fig. 5 F,G. Is it laser on/off? If so, why does the laser activate cFos expression in Cre-negative animals? Also, if these are

pyramidal neurons, why does the inhibition of the PL in pain conditions (see papers from the Neugebauer and Zamponi groups) increase cFos (see sham vs CFA data)? If these are mostly interneurons, then why does their activation cause analgesia?

Response: The authors apologize for inadequate and unclear labelling of previous panels f and g in Fig. 5. We have now corrected the error, which led to much confusion. The figure shows the following: all data shown in previous panels 5f and 5g pertain to ‘Laser On’ conditions in ChAT Cre-expressing mice (which thus express ChR2) or ChAT Cre-negative mice (which lack ChR2 expression and thus serve as negative control). The symbols denoting the individual data points are now marked clearly in the caption of revised Fig. 5f. Thus, these data, which are now assembled in revised Fig. 5f show that: (i) optogenetic stimulation of the NBM-PL pathway increases Fos⁺ neurons in the PL in baseline conditions (sham); (ii) upon CFA-induced paw inflammation, all mice show an increase in number of Fos⁺ neurons in the PL. However, optogenetic stimulation of the NBM-PL pathway further increases Fos⁺ neurons in the PL of CFA mice, i.e., over and above the Fos activation induced by inflammatory pain. We hope that these points now come across clearly in the revised results text (page 10), revised legend to Fig. 5 (pages 37-38) and in revised Fig. 5f.

The reviewer is absolutely right in pointing out that previous work suggests that the PL is deactivated in chronic pain conditions – however, to the best of our knowledge and upon perusal of the literature, we believe that this was reported in neuropathic pain models. The work of the authors the reviewer is referring to, e.g., Huang et al. (2019), pertains to neuropathic pain models. However, in inflammatory pain conditions, we do not observe a deactivation of the PL. There is a large body of evidence show that pathophysiology and mechanisms vary significantly between inflammatory and neuropathic types of chronic pain states. Our data show that NBM oscillatory activity is enhanced in inflammatory pain (Fig. 2), which is thus consistent with enhanced activity of the PL in inflammatory pain, at least in the acute stages. Thus, our data showing increase in activity of excitatory neurons in the PL upon activation of the NBM-PL pathway reveal that the NBM-PL pathway operates differently than the amygdala-PL pathway, which leads to deactivation of PL by activating inhibitory neurons in the PL (Huang et al. 2019). This is now discussed on page 17 of the revised manuscript.

Reviewer #2 (Remarks to the Author):

Summary: Oswald et al. use a multifaceted approach including optogenetics, in-vivo electrophysiology, confocal imaging, and behavioral tasks to study the role of cholinergic-GABAergic neurons within the basal nucleus of Meynert (NBM) in pain regulation. The authors present new data indicating that these NBM neurons respond to noxious stimuli with activity being enhanced by inflammatory pain. Data is presented indicating that NBM neurons play a role in attenuating pain hypersensitivity via projections to the prelimbic (PL) cortex. The authors also present data suggesting that chemogenetic activation of NBM cholinergic neurons can improve accuracy in operant conditioning tasks.

Originality and significance: Oswald et al. presents intriguing findings that address a significant knowledge gap in how a subset of forebrain cholinergic neurons play a role in pain perception and ongoing pain. While significant, concerns regarding data interpretation and approach limit enthusiasm.

Data, methodology, & statistics: The approach is valid. There are some concerns however.

Response: The authors thank the reviewer for the positive comments on originality, validity and significance of the data, and also for pointing out the need to improve clarity of presentation. We have now addressed the concerns in detail below.

1. Figure 1d does not show ChAT staining as stated in the text (Page 5, line 4 of Results). I believe this is supposed to be Figure 1c.

Response: Thank you for pointing out this error. We have now amended the text accordingly on page 5, line 8 to correctly refer to the ChAT staining shown in Fig. 1b.

2. In Figure 1F, there seems to be a single animal that showed a robust increase in theta, alpha, and beta oscillations. It is surprising that statistical significance was not reached. F and p values should be reported for these data (this was confirmed as being reported in the reporting summary associated with the manuscript). Gamma showed the smallest change, yet it was the focus of Figure 1g.

Response: In the experiments shown in Fig. 1f, statistical significance was reached for beta oscillations with weak von Frey force (non-noxious) and strong von Frey force (noxious) and for gamma oscillations for noxious intensities of force only, while theta and alpha oscillation

data were not statistically significant. We have now provided the t and p values in the legend to Fig. 1 for all frequencies of oscillations. the magnitude of change was larger in one mouse than the other mice.

It is correct that one mouse showed very large magnitude of increase in the theta and alpha frequency range, but there was nothing different about that mouse in terms of experimental design or parameters, so there is no reason to rule it out of the data set. Overall, there was high variability with the theta and alpha frequency range, and the increase in power was consistent and statistically significant in the beta and gamma frequency range despite being lower in magnitude.

We have also now amended Fig. 1g to show data on both beta and gamma oscillations.

3. It is confusing how data from Supplemental Figure 1b represents a longitudinal study design. It is showing increased stimuli force.

Response: We apologize for using the word ‘longitudinal’, which indeed does not apply to Suppl. Fig. 1b. This has now been amended in the revised text.

4. It is not clear what the authors are defining as peak mechanical hypersensitivity when referencing Figure 2c and d. Withdrawal rate % is highest at 0.6-1.0 grams of force, which is not where the largest increase in beta and gamma frequencies are observed (which is extremely variable). It seems as if the authors are defining peak mechanical hypersensitivity as the force at which there is the biggest difference in withdrawal rate between naïve and CFA animals in Supplemental Figure 1b, but this is not at all clear.

Response: We apologize for the lack of clarity of the term, and have now described more clearly what is meant with the term ‘peak mechanical hypersensitivity’. With that, we simply refer to the observation that in our hands, the largest magnitude of behavioral inflammatory hypersensitivity, as judged by withdrawal behavior in response to von Frey plantar stimulation, is reached at 24 h after intraplantar hindpaw injection of CFA. The reason why the term was mentioned in conjunction with Fig. 2c is that the oscillatory activity data shown in Fig. 2c were derived at 24 h after CFA injection.

The text has been now amended on page 6 to make this point clearer.

5. The mean differences between CFA and Naïve for both classes of neurons in Figure 3d look similar. It would be informative if full statistic values were given as it is possible that the Class 1 analysis is underpowered.

Response: The reviewer's point is well taken. We agree that the analysis of Class 1 neurons might have not been sufficiently powered to allow detecting statistically significant differences. We had initially only shown data from day 1 and day 4 post-CFA. We have now added additional data points over the period from day 2 and day 3 post-CFA and now show in total 20 data points in place of the total of 7 data points in the previous version for Class 1 units as well as a corresponding increase in the number of Class 2 units tested. Even after increasing the statistical power, the data reveal the same outcome. Thus, Class2 (fast spiking inhibitory) neurons, but not Class 1 neurons, show an increase in activity post-CFA, represented as maximal z-score in revised Fig. 3d, e. The overall analysis shown in Fig. 3c has also been amended to incorporate and collectively represent the new data points, and the inference has remained the same as before.

6. It is a bit concerning that the overlap of ChAT staining and EYFP expression in Figure 4C looks relatively low.

Response: We appreciate the reviewer's concern. In this and several previous projects in which we expressed ChR2-EYFP, we observed that the EYFP expression is most prominent in the axons and relatively less in the soma (e.g., Tan et al. 2017, Tan et al. 2019). Therefore, in the images taken, the ChAT staining, being very strong, tends to overshadow the EYFP expression. We have now provided more representative examples in revised Fig. 4c. Not all ChAT+ cells express ChR2-YFP, since the infection rate depends on the spread and titer of the viral injection.

7. The explanation of Figure 5f and 5G is lacking. The assumption is that red is laser on and black is laser off. Therefore, in the Cre- animals, there should be no expression of opsin, so it is unclear why there is an increase in Fos+ neurons. Further, it would be informative to report Fos+ neurons in L2/3 and L6 for comparison. Two-way ANOVA with Sidak's multiple comparisons tests should also be used for these comparisons.

Response: We apologize for the inadequate labelling of groups shown in former Fig. 5F, G, which led to much confusion. We have now revised this figure and the legend thoroughly, and the data are shown in revised Fig. 5f and Suppl. Fig. 4a.

The data indicate the following: all data shown pertain to ‘Laser On’ conditions in ChAT Cre-expressing mice (which thus express ChR2) or ChAT Cre-negative mice (which lack ChR2 expression and thus serve as negative control). The filled and unfilled bars refer to CFA and baseline conditions, respectively. Thus, Fig. 5f shows that: (i) optogenetic stimulation of the NBM-PL pathway increases Fos⁺ neurons in the PL in baseline conditions (sham); (ii) upon CFA-induced paw inflammation, all mice show an increase in number of Fos⁺ neurons in the PL. However, optogenetic stimulation of the NBM-PL pathway further increases Fos⁺ neurons in the PL of CFA mice, i.e., over and above the Fos activation induced by inflammatory pain. We hope that these points now come across clearly in the revised results text (page 10), revised legend to Fig. 5 (pages 37-38) and revised Fig. 5f.

Moreover, as requested by the reviewer, we have now quantified Fos⁺ cells in layers 2/3 and layer 6 and added the data to revised Suppl. Fig. 4a. We agree with the reviewer and have employed two-way ANOVA with Sidak’s multiple comparisons test for the data shown in Fig. 5f and Suppl. Fig. 4a.

Moreover, in the revised Fig. 5g,h, as requested by Rev#1, we have now also added data on the identity of Fos⁺ neurons. We immunostained pyramidal neurons in the PL for characteristic marker proteins, namely SATB2, which labels neocortical excitatory association neurons that project intra-telencephalically to other neocortical areas (Alcamo et al. 2008) and Ctip2, which labels neocortical excitatory neurons that project sub-cortically to deeper structure (Molyneaux et al. 2007). The two most abundant classes of GABAergic inhibitory interneurons were identified based upon expression of Parvalbumin (PV cells) and Somatostatin (SOM cells). Our results indicate that activation of NBM-PL projections leads to increased activity of neocortical excitatory projection neurons, but not of GABAergic inhibitory neurons.

8. The organization of Figure 6 is confusing. It looks as if CCI days 7, 11, and 28 are within 6e. Makes it difficult to follow.

Response: We agree with the reviewer, and have now rearranged the figure such that all data points related to the CCI model are clearly aligned to panel f and demarcated from other panels in revised Fig. 6.

9. The use of robustly in the following statement “heat hypersensitivity was robustly reduced in hM3D(Gq)-expressing mice as compared to mCherry-expressing” is subjective. At what point is the change not robust? Should be replaced with the word significantly.

Response: We completely agree with the reviewer and have now amended the term to ‘significantly’.

Conclusions: The robustness and validity of the study is considered adequate. Even though it is discussed briefly, there still remains some questions regarding the potential dichotomy of GABA vs. ACh effects in mPFC via release from NBM axons. Use of cholinergic receptor antagonists during experiments may have provided insight into whether behavioral changes were driven by GABA or ACh, or whether aspects of specific behaviors were controlled by GABA and ACh separately.

Response: We thank the reviewer for these concluding remarks about the robustness and validity of the study and for recognizing that the discussion section already incorporates open questions about potential dichotomy of GABA vs. Ach effects in the prefrontal cortex. It is correct that use of cholinergic receptor antagonists in combination with optogenetic modulation of the NBM-PL projections might help address a potential dichotomy. However, studies on i.c.v. injections of cholinergic receptor antagonists in vivo and in prefrontal slices ex vivo indicate that the antagonists already have strong effects of their own. This will thus obfuscate analyzing impact on effects of optogenetic stimulation of NBM-PL projections on pain behavior. Moreover, since GABA and acetylcholine are essentially co-released at the same synapses, their effects will always go hand-in-hand in any natural setting. We are therefore unsure that the experiment could lead to meaningful inferences. In line with the reviewer’s comment and Editorial advice, we have now discussed this point in more detail in the discussion section on pages 17-18.

Clarity and context: Relatively well-written. Use of subjective words like “robustly” could be reduced.

Response: We thank the reviewer for this comment, and have reduced the use of subjective words throughout the manuscript.

References cited:

Alcamo EA, *et al.* Satb2 regulates callosal projection neuron identity in the developing cerebral cortex. *Neuron* **57**, 364-377 (2008)

Huang J, *et al.* A neuronal circuit for activating descending modulation of neuropathic pain. *Nature neuroscience* **22**, 1659-1668 (2019)

Molyneaux BJ, Arlotta P, Menezes JRL, Macklis JD. Neuronal subtype specification in the cerebral cortex. *Nature Reviews Neuroscience* **8**, 427-437 (2007)

Tan LL, Oswald MJ, Heintz C, Retana Romero OA, Kaushalya SK, Monyer H, Kuner R. Gamma oscillations in somatosensory cortex recruit prefrontal and descending serotonergic pathways in aversion and nociception. *Nat Commun.* 28;10(1): 983, 2019

Tan LL, Pelzer P, Heintz C, Tang W, Gangadharan V, Flor H, Sprengel R, Kuner T, Kuner R. A pathway from midcingulate cortex to posterior insula gates nociceptive hypersensitivity. *Nat Neurosci.* 20(11):1591-1601, 2017

REVIEWER COMMENTS

Reviewer #1 (Remarks to the Author):

The results convincingly show that cholinergic modulation of the prefrontal cortex by NBM neurons provides a potent modulation of pain perception and may represent an interesting pharmacological target. The paper is interesting, well written and the authors have addressed exhaustively all the points I had raised in my previous review. I have no further comments.

Reviewer #2 (Remarks to the Author):

The authors have adequately addressed my concerns, which has improved the manuscript.

Responses to reviewers by Oswald et al.:

Reviewer #1 (Remarks to the Author):

The results convincingly show that cholinergic modulation of the prefrontal cortex by NBM neurons provides a potent modulation of pain perception and may represent an interesting pharmacological target. The paper is interesting, well written and the authors have addressed exhaustively all the points I had raised in my previous review. I have no further comments.

Response: The authors thank the reviewer for the insightful comments and supportive evaluation.

Reviewer #2 (Remarks to the Author):

The authors have adequately addressed my concerns, which has improved the manuscript.

Response: The authors are delighted that the manuscript has improved with the helpful of the constructive criticism from the reviewers.